# Interpretable Geometric Deep Learning via Learnable Randomness Injection

**Siqi Miao[1], Yunan Luo[1], Mia Liu[2], Pan Li[1,2]**
[1]Georgia Institute of Technology, [2]Purdue University
{siqi.miao,yunan,panli}@gatech.edu,{liu3173,panli}@purdue.edu

## Abstract

Point cloud data is ubiquitous in scientific fields. Recently, geometric deep learning (GDL) has been widely applied to solve prediction tasks with such data. However, GDL models are often complicated and hardly interpretable, which poses concerns to scientists who are to deploy these models in scientific analysis and experiments. This work proposes a general mechanism, *learnable randomness injection* (LRI), which allows building inherently interpretable models based on general GDL backbones. LRI-induced models, once trained, can detect the points in the point cloud data that carry information indicative of the prediction label. We also propose four datasets from real scientific applications that cover the domains of high-energy physics and biochemistry to evaluate the LRI mechanism. Compared with previous post-hoc interpretation methods, the points detected by LRI align much better and stabler with the ground-truth patterns that have actual scientific meanings. LRI is grounded by the information bottleneck principle, and thus LRI-induced models are also more robust to distribution shifts between training and test scenarios. Our code and datasets are available at `https://github.com/Graph-COM/LRI`.

## 1 Introduction

The measurement of many scientific research objects can be represented as a point cloud, i.e., a set of featured points in some geometric space. For example, in high energy physics (HEP), particles generated from collision experiments leave spacial signals on the detectors they pass through (Guest et al., 2018); In biology, a protein is often measured and represented as a collection of amino acids with spacial locations (Wang et al., 2004; 2005). Geometric quantities of such point cloud data often encode important properties of the research object, analyzing which researchers may expect to achieve new scientific discoveries (Tusnady & Simon, 1998; Aad et al., 2012).

Recently, machine learning techniques have been employed to accelerate the procedure of scientific discovery (Butler et al., 2018; Carleo et al., 2019). For geometric data as above, geometric deep learning (GDL) (Bronstein et al., 2017; 2021) has shown great promise and has been applied to the fields such as HEP (Shlomi et al., 2020; Qu & Gouskos, 2020), biochemistry (Gainza et al., 2020; Townshend et al., 2021) and so on. However, geometric data in practice is often irregular and high-dimensional. Think about a collision event in HEP that generates hundreds to thousands of particles, or a protein that consists of tens to hundreds of amino acids. Although each particle or each amino acid is located in a low-dimensional space, the whole set of points eventually is extremely irregular and high-dimensional. So, current research on GDL primarily focuses on designing neural network (NN) architectures for GDL models to deal with the above data challenge. GDL models have to preserve some symmetries of the system and incorporate the inductive biases reflected by geometric principles to guarantee their prediction quality (Cohen & Welling, 2016; Bogatskiy et al., 2020), and therefore often involve dedicated-designed complex NN architectures.

Albeit with outstanding prediction performance, the complication behind GDL models makes them hardly interpretable. However, in many scientific applications, interpretable models are in need (Roscher et al., 2020): For example, in drug discovery, compared with just predicting the binding affinity of a protein-ligand pair, it is more useful to know which groups of amino acids determine the affinity and where can be the binding site, as the obtained knowledge may guide future research directions (Gao et al., 2018; Karimi et al., 2019; 2020). Moreover, scientists tend to only trust inter-

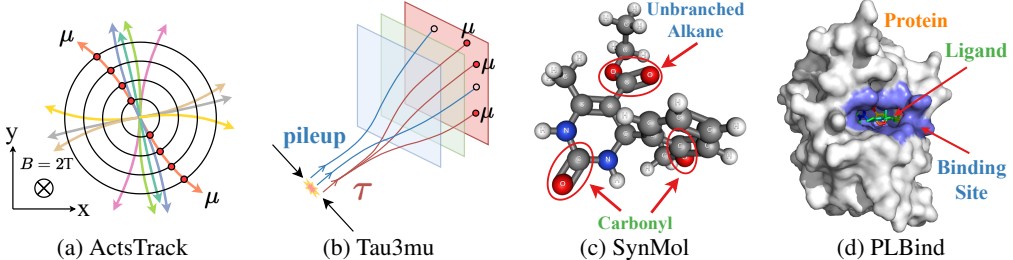

(a) ActsTrack     (b) Tau3mu     (c) SynMol     (d) PLBind

Figure 1: Illustrations of the four scientific datasets in this work to study interpretable GDL models.

pretable models in many scenarios, e.g., most applications in HEP, where data from real experiments lack labels and models have to be trained on simulation data (Nachman & Shimmin, 2019). Here, model interpretation is used to verify if a model indeed captures the patterns that match scientific principles instead of some spurious correlation between the simulation environment and labels. Unfortunately, to the best of our knowledge, there have been no studies on interpretable GDL models let alone their applications in scientific problems. Some previous post-hoc methods may be extended to interpret a pre-trained GDL model while they suffer from some limitations as to be reviewed in Sec. 2. Moreover, recent works (Rudin, 2019; Laugel et al., 2019; Bordt et al., 2022; Miao et al., 2022) have shown that the data patterns detected by post-hoc methods are often inconsistent across interpretation methods and pre-trained models, and may hardly offer reliable scientific insights.

To fill the gap, this work proposes to study interpretable GDL models. Inspired by the recent work (Miao et al., 2022), we first propose a general mechanism named *Learnable Randomness Injection* (LRI) that allows building inherently interpretable GDL models based on a broad range of GDL backbones. We then propose four datasets from real-world scientific applications in HEP and biochemistry and provide an extensive comparison between LRI-induced GDL models and previous post-hoc interpretation approaches (after being adapted to GDL models) over these datasets.

Our LRI mechanism provides model interpretation by detecting a subset of points from the point cloud that is most likely to determine the label of interest. The idea of LRI is to inject learnable randomness to each point, where, along with training the model for label prediction, injected randomness on the points that are important to prediction gets reduced. The convergent amounts of randomness on points essentially reveal the importance of the corresponding points for prediction. Specifically in GDL, as the importance of a point may be indicated by either the existence of this point in the system or its geometric location, we propose to inject two types of randomness, *Bernoulli randomness*, with the framework name *LRI-Bernoulli* to test *existence importance* of points and *Gaussian randomness* on geometric features, with the framework name *LRI-Gaussian* to test *location importance* of points. Moreover, by properly parameterized such Gaussian randomness, we may tell for a point, how in different directions perturbing its location affects the prediction result more. With such fine-grained geometric information, we may estimate the direction of the particle velocity when analyzing particle collision data in HEP. LRI is theoretically sound as it essentially uses a variational objective derived from the information bottleneck principle (Tishby et al., 2000). LRI-induced models also show better robustness to the distribution shifts between training and test scenarios, which gives scientists more confidence in applying them in practice.

We note that one obstacle to studying interpretable GDL models is the lack of valid datasets that consist of both classification labels and scientifically meaningful patterns to verify the quality of interpretation. Therefore, another significant contribution of this work is to prepare four benchmark datasets grounded on real-world scientific applications to facilitate interpretable GDL research. These datasets cover important applications in HEP and biochemistry. We illustrate the four datasets in Fig. 1 and briefly introduce them below. More detailed descriptions can be found in Appendix C.

- ActsTrack is a particle tracking dataset in HEP that is used to reconstruct the properties, such as the kinematics of a charged particle given a set of position measurements from a tracking detector. Tracking is an indispensable step in analyzing HEP experimental data as well as particle tracking used in medical applications such as proton therapy (Schulte et al., 2004; Thomson, 2013; Ai et al., 2022). *Our task* is formulated differently from traditional track reconstruction tasks: We predict the existence of a $z \to \mu\mu$ decay and use the set of points from the $\mu$'s to verify model interpretation, which can be used to reconstruct $\mu$ tracks. ActsTrack also provides a controllable environment (e.g., magnetic field strength) to study fine-grained geometric patterns.

- `Tau3Mu`, another application in HEP, is to detect a challenging signature of charged lepton flavor violating decays, i.e., the $\tau \to \mu\mu\mu$ decay, given simulated muon detector hits in proton-proton collisions. Such decays are greatly suppressed in the Standard Model (SM) of particle physics (Oerter, 2006; Blackstone et al., 2020), therefore, any detection of them is a clear signal of new physics beyond the Standard Model (Calibbi & Signorelli, 2018; Collaboration, 2021). Unfortunately, $\tau \to \mu\mu\mu$ contains particles of extremely low momentum, thus technologically impossible to trigger with traditional human-engineered algorithms. Hence, online detection with advanced models that explores the correlations between signal hits on top of background hits is required to capture such decays at the Large Hadron Collider. *Our task* is to predict the existence of $\tau \to \mu\mu\mu$ and use the detector hits left by the $\mu$'s to verify model interpretation.

- `SynMol` is a molecular property prediction task. Although some works have studied model interpretability in such tasks (McCloskey et al., 2019; Sanchez-Lengeling et al., 2020), they limit their focus on the chemical-bond-graph representations of molecules, and largely ignore their geometric features. In this work, we put focus on 3D molecular representations. *Our task* is to predict the property given by two functional groups carbonyl and unbranched alkane (McCloskey et al., 2019) and use atoms in these functional groups to verify model interpretation.

- `PLBind` is to predict protein-ligand binding affinities given the 3D structures of proteins and ligands, which is a crucial step in drug discovery, because a high affinity is one of the major drug selecting criteria (Wang & Zhang, 2017; Karimi et al., 2019). Accurately predicting their affinities with interpretable models is useful for rational drug design and may help the understanding of the underlying biophysical mechanism that enables protein-ligand binding (Held et al., 2011; Du et al., 2016; Cang & Wei, 2018). *Our task* is to predict whether the affinity is above a given threshold and use amino acids in the binding site of the test protein to verify model interpretation.

We evaluate LRI with three popular GDL backbone models DGCNN (Wang et al., 2019), Point Transformer (Zhao et al., 2021), and EGNN (Satorras et al., 2021) over the above datasets. We also extend five baseline interpretation methods to GDL for comparison. We find that interpretation results given by LRI align much better with the scientific facts than those of the baselines. Also, we observe over some datasets, LRI-Gaussian outperforms LRI-Bernoulli while on others vice versa. This implies different GDL applications may have different interpretation requirements. Effective data patterns may vary regarding how the task depends on the geometric features of the points. Interestingly, we find LRI-Gaussian can discover some fine-grained geometric patterns, such as providing high-quality estimations of the directions of particle velocities in `ActsTrack`, and a high-quality estimation of the strength of the used magnetic field. Moreover, neither of LRI mechanisms degrades the prediction performance of the used backbone models. LRI mechanisms even improve model generalization when there exist some distribution shifts from the training to test scenarios.

## 2    RELATED WORK

We review two categories of methods that can provide interpretability in the following.

**Post-hoc Interpretation Methods.** Interpretation methods falling into this category assume a pre-trained model is given and attempts to further analyze it to provide post-hoc interpretation. Among them, gradient-based methods (Zhou et al., 2016; Selvaraju et al., 2017; Sundararajan et al., 2017; Shrikumar et al., 2017; Chattopadhay et al., 2018) may be extended to interpret geometric data by checking the gradients w.r.t. the input features or intermediate embeddings of each point. Some methods to interpret graph neural networks can be applied to geometric data (Ying et al., 2019; Luo et al., 2020; Schlichtkrull et al., 2021; Yuan et al., 2021). However, these methods need to mask graph structures pre-constructed by geometric features and cannot fully evaluate the effectiveness of geometric features. Among other methods, Chen et al. (2018); Yoon et al. (2018) study pattern selection for regular data, Ribeiro et al. (2016); Lundberg & Lee (2017); Huang et al. (2022) utilize a local surrogate model, and Lundberg & Lee (2017); Chen et al. (2019b); Ancona et al. (2019); Lundberg et al. (2020) leverage the shapley value to evaluate feature importance. These methods either cannot utilize geometric features or cannot be easily applied to irregular geometric data.

**Inherently Interpretable Models.** Although vanilla attention mechanisms (Bahdanau et al., 2015; Vaswani et al., 2017) were widely used for inherent interpretability, multiple recent studies show that they cannot provide reliable interpretation, especially for data with irregular structures (Serrano & Smith, 2019; Jain & Wallace, 2019; Ying et al., 2019; Luo et al., 2020). So, some works focusing

on improving the attention mechanism for better interpretability (Bai et al., 2021; Miao et al., 2022), some propose to identify representative prototypes during training (Li et al., 2018; Chen et al., 2019a), and some methods (Taghanaki et al., 2020; Yu et al., 2021; Sun et al., 2022) adopt the information bottleneck principle (Tishby et al., 2000). However, all these methods cannot analyze geometric features in GDL. Along another line, invariant learning methods (Arjovsky et al., 2019; Chang et al., 2020; Krueger et al., 2021; Wu et al., 2022; Chen et al., 2022) focusing on out-of-distribution generalization based on causality analysis may also provide some interpretability, but these methods are typically of great complexity and cannot analyze the geometric features as well.

## 3   Preliminaries and Problem Formulation

In this section, we define some useful concepts and notations.

**GDL Tasks.** We consider a data sample is a point cloud $\mathcal{C} = (\mathcal{V}, \mathbf{X}, \mathbf{r})$, where $\mathcal{V} = \{v_1, v_2, ..., v_n\}$ is a set of $n$ points, $\mathbf{X} \in \mathbb{R}^{n \times d}$ includes $d$-dimensional features for all points, and $\mathbf{r} \in \mathbb{R}^{n \times 3}$ denotes 3D spacial coordinates of points. In this work, we introduce our notations by assuming the points are in 3D euclidean space while our method can be generalized. We focus on building a *classification* model $\hat{y} = f(\mathcal{C})$ to predict the class label $y$ of $\mathcal{C}$. Regression tasks are left for future studies.

**GDL Models.** The *first* class of DGL models view each sample of points as an unordered set. It learns a dense representation $\mathbf{z}_v$ for each $v \in \mathcal{V}$, and then applies a permutation invariant function, e.g., sum/mean/max pooling, to aggregate all point representations so that they can handle irregular data (Zaheer et al., 2017; Charles et al., 2017). The *second* class of methods can better utilize geometric features and local information. These methods first construct a $k$-nn graph $\mathcal{G}$ over the points in each sample based on their distances, e.g., $\|\mathbf{r}_v - \mathbf{r}_u\|$, and iteratively update the representation of point $v$ via aggregation $\text{AGG}(\{\mathbf{z}_u \mid u \in \mathcal{N}(v)\})$, where $\mathcal{N}(v)$ is the neighbours of point $v$ in graph $\mathcal{G}$ and AGG is a permutation invariant function. Then, another function is used to aggregate all point representations to make predictions. Compared with graph neural networks (GNNs) (Kipf & Welling, 2017; Xu et al., 2019; Veličković et al., 2018) that encode graph-structured data without geometric features, GDL models often process geometric features carefully: Typically, these features are transformed into some scalars such as distances, angles and used as features so that some group (e.g., $E(3)$, $SO(3)$) invariances of the prediction can be kept (Fuchs et al., 2020; Gasteiger et al., 2020; Beaini et al., 2021; Satorras et al., 2021; Schütt et al., 2021). Some models that perform 3D convolution over 3D data also belong to the second class because the convolution kernels can be viewed as one way to define the distance scalars for graph construction and neighborhood aggregation (Schütt et al., 2017; Thomas et al., 2018; Wu et al., 2019). The *third* class will dynamically construct the $k$-nn graphs based on the hidden representations of points (Wang et al., 2019; Qu & Gouskos, 2020; Zhao et al., 2021). In this work, we focus on using the second class of models' architectures as the backbones because most scientific applications adopt this class of models.

**Interpretable Patterns in GDL.** Given a sample $\mathcal{C} = (\mathcal{V}, \mathbf{X}, \mathbf{r})$, our goal of building an inherently interpretable model is that the model by itself can identify a subset of points $\mathcal{C}_s = (\mathcal{V}_s, \mathbf{X}_s, \mathbf{r}_s)$ that best indicates the label $y$. Mostly, $\mathcal{C}_s$ will have a scientific meaning. For example, in the task to detect $\tau \to \mu\mu\mu$ decay, our model should identify the detector hits left by the three $\mu$'s but not the hits from other particles. We consider two types of indications of the label given by a point: *existence importance*, i.e., whether the point exists in the cloud is important to determine the label, and *location importance*, i.e. whether the geometric location of the point is important to determine the label. In the above example, the existence of the detector hits left by the three $\mu$'s is of course important. On the other hand, the locations of these hits are also crucial because location features reflect the momentum of these particles when they pass through detectors, which should satisfy equations regarding certain invariant mass if they are indeed generated from a $\tau \to \mu\mu\mu$ decay.

## 4   Methodology

In this section, we introduce our method *Learnable Randomness Injection* (LRI). In LRI, we have an interpreter $g$ and a classifier $f$. $g$ is to encode the original data and generate randomness to perturb the data. $f$ is to encode the perturbed data and make predictions. $g$ and $f$ are trained together to make accurate predictions while providing interpretability. LRI can be applied to a large class of GDL models to make them interpretable. We may choose a GDL model architecture as the backbone to build the data encoders in $g$ and $f$. These encoders could share or not share parameters. Below,

Figure 2: The architectures of LRI-Bernoulli (top) and LRI-Gaussian (bottom).

we introduce specifics about this procedure. We first describe LRI-Bernoulli, where Bernoulli randomness is injected to measure the *existence important* of points. Then, we introduce LRI-Gaussian, which injects Gaussian randomness into geometric features to test the *location importance* of points. Finally, we connect our objectives with the information bottleneck principle (Tishby et al., 2000).

## 4.1 LRI-BERNOULLI TO TEST EXISTENCE IMPORTANCE

**Pipeline.** Given a sample $\mathcal{C} = (\mathcal{V}, \mathbf{X}, \mathbf{r})$, we first construct a $k$-nn graph $\mathcal{G}$ based on the euclidean distance $\|\mathbf{r}_v - \mathbf{r}_u\|$ between every pair of points $v, u \in \mathcal{V}$. As shown in the top of Fig. 2, the interpreter $g$ encodes $\mathcal{C}$, generates a representation $\mathbf{z}_v$ for each point $v$ and uses the last component $h$ to map $\mathbf{z}_v$ to $p_v \in [0, 1]$. Here, $h$ consists of an MLP plus a sigmoid layer, and samples a Bernoulli mask for each point via $m_v \sim \text{Bern}(p_v)$. The sampling is based on a reparameterization trick (Jang et al., 2017; Maddison et al., 2017) to make $\frac{dm_v}{dp_v}$ computable. The perturbed data $\tilde{\mathcal{C}}$ is yielded by removing the points with $m_v = 0$ in $\mathcal{C}$. The edges in $\mathcal{G}$ connected to these points are also masked and removed, which gives a graph $\tilde{\mathcal{G}}$. Finally, the classifier $f$ takes as inputs $\tilde{\mathcal{C}}$ and $\tilde{\mathcal{G}}$ to make predictions.

**Objective.** Eq. 1 shows our objective for each sample $\mathcal{C}$, where the first term is a cross-entropy loss for classification and the second term is a KL divergence regularizer. $\beta$ is the regularization coefficient and $\text{Bern}(\alpha)$ is a predefined Bernoulli distribution with hyperparameter $\alpha < 1$.

$$\min \mathcal{L}_{CE}(f(\tilde{\mathcal{C}}, \tilde{\mathcal{G}}), y) + \beta \sum_{v \in \mathcal{V}} D_{\text{KL}}(\text{Bern}(p_v)\|\text{Bern}(\alpha)). \tag{1}$$

Here, $f$ is optimized via the first term. The interpreter $g$ is optimized via the gradients that pass through the masks $\{m_v\}_{v \in \mathcal{V}}$ contained in $\tilde{\mathcal{C}}$ and $\tilde{\mathcal{G}}$ in the first term, and $\{p_v\}_{v \in \mathcal{V}}$ in the second term.

**Interpretation Rationale.** The interpretation is given by the competition between the two terms in Eq. 1. The first term is to achieve good classification performance so it tends to denoise the data $\tilde{\mathcal{C}}$ by reducing the randomness generated by $g$, i.e., $p_v \to 1$. The second term, on the other hand, tends to keep the level of randomness, i.e., $p_v \to \alpha$. After training, $p_v$ will converge to some value. If the existence of a point $v$ is important to the label $y$, then $p_v$ should be close to 1. Otherwise, $p_v$ is close to $\alpha$. We use $p_v$'s to rank the points $v \in \mathcal{V}$ according to their existence importance.

## 4.2 LRI-GAUSSIAN TO TEST LOCATION IMPORTANCE

**Pipeline.** We start from the same graph $\mathcal{G}$ as LRI-Bernoulli. As shown in the bottom of Fig. 2, here, the interpreter $g$ will encode the data and map it to a covariance matrix $\mathbf{\Sigma}_v \in \mathbb{R}^{3 \times 3}$ for each point $v$. Gaussian randomness $\epsilon_v \sim \mathcal{N}(\mathbf{0}, \mathbf{\Sigma}_v)$ is then sampled to perturb the geometric features $\tilde{\mathbf{r}}_v = \mathbf{r}_v + \epsilon_v$ of $v$. Note that, to test location importance, a new $k$-nn graph $\tilde{\mathcal{G}}$ is constructed based on perturbed distances $\|\tilde{\mathbf{r}}_v - \tilde{\mathbf{r}}_u\|$. Reconstructing $\tilde{\mathcal{G}}$ is necessary because using the original graph $\mathcal{G}$ will leak information from the original geometric features $\mathbf{r}$. Finally, the classifier $f$ takes as inputs the location-perturbed data $\tilde{\mathcal{C}} = (\mathcal{V}, \mathbf{X}, \tilde{\mathbf{r}})$ and $\tilde{\mathcal{G}}$ to make predictions.

**Objective.** Eq. 2 shows the objective of LRI-Gaussian for each sample $\mathcal{C}$. Different from Eq. 1, here the regularization is a Gaussian $\mathcal{N}(\mathbf{0}, \sigma\mathbf{I})$, where $\mathbf{I}$ is an identity matrix and $\sigma$ is a hyperparameter.

$$\min \mathcal{L}_{CE}(f(\tilde{\mathcal{C}}, \tilde{\mathcal{G}}), y) + \beta \sum_{v \in \mathcal{V}} D_{\text{KL}}(\mathcal{N}(\mathbf{0}, \mathbf{\Sigma}_v)\|\mathcal{N}(\mathbf{0}, \sigma\mathbf{I})). \tag{2}$$

Again, the classifier $f$ will be optimized via the first term. The interpreter $g$ will be optimized via the gradients that pass through the perturbation $\{\epsilon_v\}_{v \in \mathcal{V}}$ implicitly contained in $\tilde{\mathcal{C}}$ and $\tilde{\mathcal{G}}$ in the first term, and $\{\mathbf{\Sigma}_v\}_{v \in \mathcal{V}}$ in the second term. However, there are two technical difficulties to be addressed.

Table 1: Statistics of the four datasets.

| | # Classes | # Features in $\mathbf{X}$ | # Dimensions in $\mathbf{r}$ | # Samples | Avg. # Points/Sample | Avg. # Important Points/Sample | Class Ratio | Split Scheme | Split Ratio |
|---|---|---|---|---|---|---|---|---|---|
| ActsTrack | 2 | 0 | 3 | 3241 | 109.1 | 22.8 | 39/61 | Random | 70/15/15 |
| Tau3Mu | 2 | 1 | 2 | 129687 | 16.9 | 5.5 | 24/76 | Random | 70/15/15 |
| SynMol | 2 | 1 | 3 | 8663 | 21.9 | 6.6 | 18/82 | Patterns | 78/11/11 |
| PLBind | 2 | 3 | 3 | 10891 | 339.8 | 132.2 | 29/71 | Time | 92/6/2 |

First, how to parameterize $\boldsymbol{\Sigma}_v$ as it should be positive definite, and then how to make $\frac{d\epsilon_v}{d\boldsymbol{\Sigma}_v}$ computable? Our solution is to let the last component $h$ in $g$ map representation $\mathbf{z}_v$ not to $\boldsymbol{\Sigma}_v$ directly but to a dense matrix $\mathbf{U}_v \in \mathbb{R}^{3\times3}$ via an MLP and two scalars $a_1, a_2 \in \mathbb{R}^+$ via a softplus layer. Then, the covariance matrix is computed by $\boldsymbol{\Sigma}_v = a_1 \mathbf{U}_v \mathbf{U}_v^T + a_2 \mathbf{I}$. Moreover, we find using $\boldsymbol{\Sigma}_v$ and the reparameterization trick for multivariate Gaussian implemented by PyTorch is numerically unstable as it includes Cholesky decomposition. So, instead, we use the reparameterization trick $\epsilon_v = \sqrt{a_1}\mathbf{U}_v\mathbf{s}_1 + \sqrt{a_2}\mathbf{I}\mathbf{s}_2$, where $\mathbf{s}_1, \mathbf{s}_2 \sim \mathcal{N}(\mathbf{0}, \mathbf{I})$. It is not hard to show that $\mathbb{E}[\epsilon_v\epsilon_v^T] = \boldsymbol{\Sigma}_v$.

Second, the construction of the $k$-nn graph $\tilde{\mathcal{G}}$ based on $\tilde{\mathbf{r}}_v$ is not differentiable, which makes the gradients of $\{\epsilon_v\}_{v\in\mathcal{V}}$ that pass through the structure of $\tilde{\mathcal{G}}$ not computable. We address this issue by associating each edge $v, u$ in $\tilde{\mathcal{G}}$ with a weight $w_{vu} \in (0, 1)$ that monotonically decreases w.r.t. the distance, $w_{vu} = \phi(\|\tilde{\mathbf{r}}_v - \tilde{\mathbf{r}}_u\|)$. These weights are used in the neighborhood aggregation procedure in $f$. Specifically, for the central point $v$, $f$ adopts aggregation $\text{AGG}(\{w_{vu}\mathbf{z}_u \mid u \in \mathcal{N}(v)\})$, where $\mathbf{z}_u$ is the representation of the neighbor point $u$ in the current layer. This design makes the structure of $\tilde{\mathcal{G}}$ differentiable. Moreover, because we set $w_{uv} < 1$, the number of used nearest neighbors to construct $\tilde{\mathcal{G}}$ is "conceptually" smaller than $k$. So, in practice, we choose a slightly larger number (say $1.5k$) of nearest neighbors to construct $\tilde{\mathcal{G}}$ and adopt the above strategy.

**Interpretation Rationale.** The interpretation rationale is similar to that of LRI-Bernoulli, i.e., given by the competition between the two terms in Eq. 1. The first term is to achieve good classification performance by reducing the randomness generated by $g$. The second term, on the other hand, tends to keep the level of randomness, i.e., $\boldsymbol{\Sigma}_v \to \sigma\mathbf{I}$. After training, the convergent determinant $|\boldsymbol{\Sigma}_v|$ which characterizes the entropy of injected Gaussian randomness, indicates the *location importance* of point $v$. We use $|\boldsymbol{\Sigma}_v|$'s to rank the points $v \in \mathcal{V}$ according to their location importance.

**Fine-grained Interpretation on Location Importance.** Interestingly, the convergent $\boldsymbol{\Sigma}_v$ implies more fine-grained geometric information, i.e., how different directions of perturbations on point $v$ affect the prediction. This can be analyzed by checking the eigenvectors of $\boldsymbol{\Sigma}_v$. As illustrated in the figure on the right, $\boldsymbol{\Sigma}_v$ of point $v$ at $A$ is represented by the ellipses $\{\mathbf{x} : \mathbf{x}^T\boldsymbol{\Sigma}_v\mathbf{x} < \theta\}$ for different $\theta$'s. It tells perturbing $v$ towards the direction $B$ affects the prediction less than perturbing $v$ towards the orthogonal direction. As a showcase, later, we use such fine-grained information to conduct an in-depth analysis of HEP data.

### 4.3 Connecting LRI and the Information Bottleneck Principle.

Our objectives Eq. 1 and Eq. 2 are essentially variational upper bounds of the information bottleneck (IB) principle (Tishby et al., 2000; Alemi et al., 2017) whose goal is to reduce the mutual information between $\mathcal{C}$ and $\tilde{\mathcal{C}}$ while keeping the mutual information between $\tilde{\mathcal{C}}$ and the label, i.e., $\min -I(\tilde{\mathcal{C}}; Y) + \beta I(\tilde{\mathcal{C}}; \mathcal{C})$. We provide derivations in Appendix A. Grounded on the IB principle, LRI tends to extract minimal sufficient information to make predictions and can be more robust to distribution shifts between training and test datasets (Achille & Soatto, 2018; Wu et al., 2020; Miao et al., 2022).

## 5 Benchmarking Interpretable GDL

In this section, we will evaluate LRI-Bernoulli, LRI-Gaussian and some baseline methods extended to GDL over the proposed four datasets. Here, we briefly describe our setup and put more details on the datasets, hyperparameter tuning, and method implementations, in Appendix C, D, and E, respectively. A summary of dataset statistics is shown in Table 1.

**Backbone Models** include DGCNN (Wang et al., 2019), Point Transformer (Zhao et al., 2021), and EGNN (Satorras et al., 2021) which have been widely used for scientific GDL (Qu & Gouskos, 2020; Atz et al., 2021; Gagliardi et al., 2022).

**Baseline interpretation methods** include three masking-based methods BernMask, BernMask-P and PointMask, and two gradient-based methods GradGeo and GradGAM. Masking-based methods

Table 2: Interpretation performance on the four datasets. The **Bold**[†], **Bold**, and Underline highlight the first, second, and third best results, respectively. All results are reported with mean $\pm$ std.

| ActsTrack | DGCNN | | | Point Transformer | | | EGNN | | |
|---|---|---|---|---|---|---|---|---|---|
| | ROC AUC | Prec@12 | Prec@24 | ROC AUC | Prec@12 | Prec@24 | ROC AUC | Prec@12 | Prec@24 |
| Random | 50 | 21 | 21 | 50 | 21 | 21 | 50 | 21 | 21 |
| GradGeo | 65.92 ± 1.55 | 30.71 ± 1.46 | 30.20 ± 1.07 | 65.92 ± 1.61 | 31.76 ± 1.04 | 30.06 ± 1.32 | 67.57 ± 0.65 | 31.09 ± 1.37 | 30.87 ± 1.24 |
| BernMask | 68.85 ± 4.72 | 45.90 ± 7.14 | 42.72 ± 7.91 | 76.94 ± 1.99 | 71.17 ± 1.66 | 57.10 ± 3.22 | 50.94 ± 3.86 | 18.67 ± 2.06 | 20.00 ± 3.13 |
| BernMask-P | 88.16 ± 3.22 | 76.94 ± 11.30 | 66.71 ± 6.99 | 84.36 ± 2.64 | 73.24 ± 6.60 | 60.41 ± 5.44 | 30.81 ± 27.63 | 17.38 ± 32.08 | 13.92 ± 21.22 |
| PointMask | 49.85 ± 6.11 | 21.05 ± 4.67 | 21.54 ± 4.16 | 50.66 ± 0.91 | 22.43 ± 5.78 | 20.63 ± 3.63 | 49.55 ± 1.40 | 20.35 ± 1.14 | 20.06 ± 1.01 |
| GradGAM | 82.96 ± 2.42 | 84.29 ± 2.25 | 66.34 ± 2.56 | 83.07 ± 2.28 | 82.13 ± 0.92 | 64.64 ± 1.89 | 79.44 ± 2.62 | 78.38 ± 1.96 | 52.55 ± 3.38 |
| LRI-Bernoulli | 90.29 ± 1.39 | 89.36 ± 1.20 | 74.59 ± 1.28 | 87.06 ± 2.49 | 85.71 ± 0.99 | 67.65 ± 1.22 | 78.57 ± 3.34 | 81.56 ± 1.13 | 55.29 ± 1.99 |
| LRI-Gaussian | 95.49[†] ± 0.34 | 92.40[†] ± 0.64 | 79.87[†] ± 0.72 | 93.11[†] ± 1.50 | 88.39[†] ± 4.36 | 74.71[†] ± 1.99 | 94.34[†] ± 0.70 | 91.54[†] ± 1.49 | 76.78[†] ± 0.81 |

| Tau3Mu | DGCNN | | | Point Transformer | | | EGNN | | |
|---|---|---|---|---|---|---|---|---|---|
| | ROC AUC | Prec@3 | Prec@7 | ROC AUC | Prec@3 | Prec@7 | ROC AUC | Prec@3 | Prec@7 |
| Random | 50 | 35 | 35 | 50 | 35 | 35 | 50 | 35 | 35 |
| GradGeo | 80.76 ± 0.31 | 68.86 ± 0.37 | 56.05 ± 0.34 | 79.62 ± 0.33 | 67.96 ± 0.49 | 55.13 ± 0.24 | 78.05 ± 1.13 | 64.75 ± 2.09 | 54.28 ± 0.78 |
| BernMask | 54.28 ± 1.15 | 50.88 ± 1.64 | 42.21 ± 0.94 | 30.00 ± 0.40 | 12.48 ± 0.58 | 18.81 ± 0.27 | 72.27 ± 2.66 | 61.27 ± 3.68 | 51.91 ± 1.63 |
| BernMask-P | 54.99 ± 14.06 | 43.56 ± 13.59 | 39.23 ± 9.49 | 76.46 ± 4.05 | 59.39 ± 4.78 | 53.41 ± 3.30 | 70.99 ± 5.59 | 56.36 ± 6.04 | 50.06 ± 4.44 |
| PointMask | 52.92 ± 2.56 | 39.55 ± 2.27 | 36.71 ± 1.48 | 57.33 ± 2.28 | 44.00 ± 1.91 | 38.53 ± 2.41 | 55.90 ± 5.39 | 39.92 ± 8.20 | 37.05 ± 4.74 |
| GradGAM | 68.42 ± 4.04 | 51.82 ± 6.02 | 45.78 ± 3.60 | 80.91[†] ± 0.35 | 61.14 ± 1.46 | 54.36 ± 0.67 | 75.84 ± 1.45 | 62.03 ± 2.41 | 53.19 ± 1.32 |
| LRI-Bernoulli | 77.88 ± 1.03 | 70.66 ± 0.90 | 55.94 ± 0.77 | 77.72 ± 1.52 | 67.73 ± 2.59 | 55.74 ± 1.15 | 78.71 ± 0.66 | 65.99 ± 0.84 | 55.98 ± 0.57 |
| LRI-Gaussian | 81.38[†] ± 0.62 | 73.13[†] ± 1.10 | 58.28[†] ± 0.59 | 79.58 ± 0.66 | 70.32[†] ± 0.76 | 57.05[†] ± 0.53 | 80.02[†] ± 0.39 | 71.20[†] ± 0.93 | 57.07 ± 0.41 |

| SynMol | DGCNN | | | Point Transformer | | | EGNN | | |
|---|---|---|---|---|---|---|---|---|---|
| | ROC AUC | Prec@5 | Prec@8 | ROC AUC | Prec@5 | Prec@8 | ROC AUC | Prec@5 | Prec@8 |
| Random | 50 | 31 | 31 | 50 | 31 | 31 | 50 | 31 | 31 |
| GradGeo | 72.10 ± 9.66 | 59.59 ± 11.05 | 50.30 ± 8.70 | 76.94 ± 1.43 | 62.30 ± 0.78 | 55.29 ± 0.87 | 73.49 ± 5.23 | 61.85 ± 5.26 | 50.46 ± 3.95 |
| BernMask | 49.69 ± 9.22 | 34.37 ± 9.32 | 32.15 ± 7.64 | 25.28 ± 3.52 | 6.85 ± 1.57 | 8.65 ± 1.14 | 59.76 ± 9.09 | 49.96 ± 7.56 | 40.72 ± 7.25 |
| BernMask-P | 70.51 ± 39.52 | 63.02 ± 36.13 | 52.93 ± 30.14 | 87.23 ± 6.07 | 75.39 ± 9.74 | 63.00 ± 7.31 | 90.00 ± 7.85 | 85.52 ± 5.75 | 68.94 ± 6.37 |
| PointMask | 74.22 ± 3.31 | 71.54 ± 4.27 | 55.18 ± 2.86 | 72.03 ± 2.10 | 60.13 ± 2.57 | 51.11 ± 1.43 | 65.43 ± 6.63 | 53.89 ± 2.25 | 48.11 ± 3.05 |
| GradGAM | 81.98 ± 5.54 | 78.80 ± 6.67 | 59.86 ± 5.86 | 85.54 ± 1.19 | 80.24 ± 1.98 | 64.38 ± 1.49 | 57.00 ± 5.52 | 48.07 ± 8.56 | 41.30 ± 5.08 |
| LRI-Bernoulli | 96.03 ± 1.54 | 87.11 ± 4.51 | 74.57 ± 1.57 | 91.69 ± 1.52 | 82.72 ± 2.20 | 68.37 ± 1.11 | 90.64 ± 3.30 | 71.96 ± 5.97 | 68.08 ± 4.18 |
| LRI-Gaussian | 99.02[†] ± 0.36 | 97.72[†] ± 0.94 | 77.04[†] ± 0.43 | 95.35[†] ± 1.02 | 87.09[†] ± 1.97 | 72.26[†] ± 1.40 | 97.28[†] ± 0.65 | 91.52[†] ± 1.28 | 74.05[†] ± 1.18 |

| PLBind | DGCNN | | | Point Transformer | | | EGNN | | |
|---|---|---|---|---|---|---|---|---|---|
| | ROC AUC | Prec@20 | Prec@40 | ROC AUC | Prec@20 | Prec@40 | ROC AUC | Prec@20 | Prec@40 |
| Random | 50 | 45 | 45 | 50 | 45 | 45 | 50 | 45 | 45 |
| GradGeo | 52.83 ± 4.63 | 55.68 ± 2.47 | 53.79 ± 1.83 | 58.68 ± 2.83 | 59.30 ± 3.13 | 57.85 ± 3.57 | 57.78[†] ± 2.61 | 61.00 ± 2.24 | 60.11 ± 1.76 |
| BernMask | 48.18 ± 4.14 | 48.36 ± 3.32 | 48.00 ± 3.40 | 59.73[†] ± 2.33 | 59.30 ± 3.09 | 58.73 ± 2.90 | 49.83 ± 2.17 | 40.34 ± 3.96 | 41.99 ± 2.32 |
| BernMask-P | 48.88 ± 5.66 | 42.70 ± 8.37 | 42.46 ± 7.88 | 56.47 ± 2.77 | 56.86 ± 3.79 | 54.53 ± 4.64 | 51.96 ± 6.80 | 60.68 ± 7.95 | 57.69 ± 6.41 |
| PointMask | 51.38 ± 3.12 | 45.36 ± 1.91 | 45.22 ± 1.52 | 52.92 ± 3.83 | 44.34 ± 3.50 | 44.50 ± 4.52 | 50.00 ± 0.00 | 45.10 ± 0.00 | 45.00 ± 0.00 |
| GradGAM | 53.76 ± 3.38 | 55.50 ± 4.83 | 54.23 ± 4.42 | 56.51 ± 3.32 | 56.54 ± 6.59 | 54.46 ± 5.65 | 49.73 ± 2.18 | 56.92 ± 6.63 | 53.96 ± 3.53 |
| LRI-Bernoulli | 55.47[†] ± 2.06 | 67.56[†] ± 5.38 | 61.02[†] ± 5.68 | 59.53 ± 1.94 | 72.98[†] ± 3.85 | 67.33[†] ± 2.08 | 57.07 ± 3.09 | 73.22[†] ± 2.32 | 66.89 ± 2.07 |
| LRI-Gaussian | 51.81 ± 3.24 | 63.88 ± 3.18 | 60.37 ± 3.09 | 54.05 ± 2.94 | 62.76 ± 5.93 | 60.44 ± 5.62 | 50.32 ± 3.80 | 72.64 ± 2.04 | 69.40[†] ± 1.44 |

attempt to learn a mask $\in [0, 1]$ for each point and may help to test existence importance of points. Among them, BernMask and BernMask-P are post-hoc extended from two previous methods on graph-structured data, i.e., Ying et al. (2019) and Luo et al. (2020). BernMask and BernMask-P differ in the way they generate the masks, where BernMask-P utilizes a parameterized mask generator while BernMask optimizes a randomly initialized mask with no other parameterization. PointMask is an inherently interpretable model adopted from Taghanaki et al. (2020). Thus, they are the baselines of LRI-Bernoulli. GradGeo is extended from Shrikumar et al. (2017) and checks the gradients w.r.t. geometric features, which may help with testing location importance, and thus is a baseline of LRI-Gaussian. GradGAM is extended from Selvaraju et al. (2017) and leverages the gradients w.r.t. the learned representations of points, which also reflects the importance of points to the prediction.

**Metrics.** On each dataset, the model is trained based on the prediction (binary classification) task. Then, we compare interpretation labels ($\in \{0, 1\}$) of points with the learned point importance scores to measure interpretation performance. We report two metrics: interpretation ROC AUC and precision@$m$. Beyond interpretation, the model prediction performance is also measured in ROC AUC and reported in Sec. 5.5, which is to make sure that all models are pre-trained well for those post-hoc baselines and to verify that LRI-induced models also have good prediction accuracy.

**Hyperparameter Tuning.** Gradient-based methods have no hyperparameters to tune, while all other methods are tuned based on the validation prediction performance for a fair comparison, because tuning on interpretation performance is impossible in a real-world setting. For LRI-Bernoulli, $\alpha$ is tuned in $\{0.5, 0.7\}$; for LRI-Gaussian, as the geometric features $\mathbf{r}$ in our datasets are all centered at the origin, $\sigma$ is tuned based on the 5th and 10th percentiles of entries of abs($\mathbf{r}$). And, $\beta$ is tuned in $\{1.0, 0.1, 0.01\}$ for both methods after normalization based on the total number of points.

## 5.1 ActsTrack: End-to-end Pattern Recognition and Track Reconstruction

Here, we are to predict whether a collision event contains a $z \to \mu\mu$ decay. Each point in a point cloud sample is a detector hit where a particle passes through. We use the hits from the decay to

Table 3: The angle (°, mean±std) between the velocity and the first principal component of $\Sigma_v$ (LRI-Gaussian) v.s. between the velocity and the direction orthogonal to the gradient of $\mathbf{r}_v$ in x-y space (GradGeo) under different magnetic field strengths (T).

| | 2T | 4T | 6T | 8T | 10T | 12T | 14T | 16T | 18T | 20T |
|---|---|---|---|---|---|---|---|---|---|---|
| Random | 45 | 45 | 45 | 45 | 45 | 45 | 45 | 45 | 45 | 45 |
| GradGeo | 22.89 ± 2.45 | 18.85 ± 1.29 | 34.79 ± 2.12 | 30.57 ± 2.82 | 30.27 ± 2.54 | 37.22 ± 3.18 | 38.01 ± 3.24 | 39.23 ± 1.60 | 38.01 ± 3.89 | 36.26 ± 5.34 |
| LRI-Gaussian | 5.09 ± 0.97 | 5.17 ± 0.42 | 5.65 ± 1.05 | 6.67 ± 1.17 | 5.99 ± 1.71 | 6.66 ± 1.25 | 7.50 ± 2.37 | 7.19 ± 1.00 | 7.57 ± 1.20 | 7.89 ± 1.03 |

test interpretation performance. We evaluate all methods on the data generated with a magnetic field $B = 2T$ parallel to the z axis (see Fig. 1a). Table 2 shows the results, where LRI-Gaussian works consistently the best on all backbones and metrics, and LRI-Bernoulli achieves the second best performance. This indicates that both the *existence* and *locations* of those detector hits left by the $\mu$'s are important to predictions. GradGAM is also a competitive baseline, which even outperforms both BernMask and BernMask-P. While BernMask-P performs decently on two backbones, it fails to provide interpretability for EGNN and its results have high variances, probably due to the unstable issue of post-hoc interpretations, as described in Miao et al. (2022). BernMask, PointMask, and GradGeo seem unable to provide valid interpretability for `ActsTrack`.

**Fine-grained Geometric Patterns.** We find LRI-Gaussian can discover fine-grained geometric patterns. Intuitively, slight perturbation of each point along the underlying track direction will affects the model prediction less than the same level of perturbation orthogonal to the track. Therefore, the principal component of $\Sigma_v$ in x-y space, i.e., the space orthogonal to the direction of the background magnetic field, can give an estimation of the track direction at $v$. Table 3 provides the evaluation of track direction (velocity direction) estimation based on analyzing $\Sigma_v$. Here, we test the background magnetic field changing from 2T to 20T. LRI-Gaussian is far more accurate than GradGeo. The latter uses the gradients of different coordinates to compute the most sensitive direction. Moreover, the ratio between the lengths

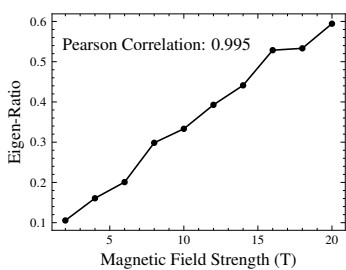

Figure 3: Eigen-ratio of $\Sigma_v$ v.s. magnetic field strength $B$ (T).

of two principal components of $\Sigma_v$ in x-y space gives an estimation of the curvature of the track at $v$, which is proportional to the strength of the magnetic field $B$ up to a constant multiplier (due to the law – Lorentz force $F \propto B$). Therefore, we can estimate $B$ by analyzing $\Sigma_v$. Fig. 3 shows this approach provides an almost accurate estimation of $B$ up to a constant multiplier. Fig. 4 provides visualizations of the yielded fine-grained patterns, and a detailed analysis of the obtained fine-grained interpretation patterns can be found in Appendix B.

## 5.2 TAU3MU: $\tau \to \mu\mu\mu$ DECAY DETECTION IN PROTON-PROTON COLLISIONS

This task is to predict whether a proton-proton collision event contains a $\tau \to \mu\mu\mu$ decay, which is similar to `ActsTrack`, while in `Tau3Mu` the $\mu$'s are a lot softer, and only the hits from the first layer of detectors are used. Each point in a point cloud sample is a detector hit associated with a local bending angle and a 2D coordinate in the pseudorapidity-azimuth ($\eta$-$\phi$) space. We use the hits generated from this decay to test interpretation performance. As shown in Table 2, LRI-Gaussian still works the best. LRI-Bernoulli and GradGeo are close, and both are the second best. While GradGAM still works well on some backbones, all masking-based methods do not perform well.

## 5.3 SYNMOL: MOLECULAR PROPERTY PREDICTION WITH SYNTHETIC PROPERTIES

This task is to predict whether a molecule contains both functional groups *branched alkanes* and *carbonyl*, which together give certain synthetic properties (McCloskey et al., 2019; Sanchez-Lengeling et al., 2020). Each point in a point cloud sample is an atom associated with a 3D coordinate and a categorical feature indicating the atom type. We use the atoms in these two functional groups to test interpretation performance. As shown in Table 2, LRI-Gaussian performs consistently the best by only perturbing geometric features in molecules, and LRI-Bernoulli works the second best and achieves comparable performance with LRI-Gaussian on Point Transformer. This shows that both the existence and locations of atoms are critical and further validates the benefit of using 3D representations of molecules in the tasks like molecular property prediction. Among other methods,

Table 4: Generalization performance of LRI. Classification AUC on test sets are reported with mean±std for all backbones on the four datasets.

| | ActsTrack | | | Tau3Mu | | | SynMol | | | PLBind | | |
|---|---|---|---|---|---|---|---|---|---|---|---|---|
| | DGCNN | Point Transformer | EGNN | DGCNN | Point Transformer | EGNN | DGCNN | Point Transformer | EGNN | DGCNN | Point Transformer | EGNN |
| ERM | $97.99 \pm 0.38$ | $96.75 \pm 0.21$ | $97.45 \pm 0.52$ | $87.38 \pm 0.08$ | $86.20 \pm 0.13$ | $86.45 \pm 0.09$ | $99.95 \pm 0.03$ | $98.14 \pm 0.42$ | $99.87 \pm 0.05$ | $80.17 \pm 5.23$ | $83.13 \pm 1.19$ | $86.80 \pm 3.52$ |
| LRI-Bernoulli | $98.11 \pm 0.09$ | $97.39 \pm 0.27$ | $98.52 \pm 0.35$ | $87.45 \pm 0.06$ | $86.44 \pm 0.08$ | $86.56 \pm 0.11$ | $99.82 \pm 0.15$ | $97.93 \pm 0.46$ | $99.81 \pm 0.05$ | $81.25 \pm 2.01$ | $86.83 \pm 2.06$ | $86.93 \pm 3.91$ |
| LRI-Gaussian | $98.17 \pm 0.24$ | $98.04 \pm 0.54$ | $98.85 \pm 0.14$ | $87.74 \pm 0.15$ | $86.03 \pm 0.37$ | $86.67 \pm 0.08$ | $99.98 \pm 0.01$ | $98.80 \pm 0.31$ | $99.88 \pm 0.05$ | $84.34 \pm 5.32$ | $86.71 \pm 0.64$ | $87.53 \pm 0.95$ |

Table 5: Generalization performance of LRI with distribution shifts. The column name $d_1$-$d_2$ denotes the models are validated and tested on samples with $d_1$ and $d_2$ tracks, respectively.

| ActsTrack | 10-10 | 15-20 | 20-30 | 25-40 | 30-50 | 35-60 | 40-70 |
|---|---|---|---|---|---|---|---|
| ERM | $96.33 \pm 0.65$ | $93.83 \pm 0.34$ | $91.14 \pm 1.07$ | $87.85 \pm 1.12$ | $85.96 \pm 1.74$ | $84.19 \pm 1.54$ | $82.87 \pm 0.74$ |
| LRI-Bernoulli | $97.05 \pm 0.71$ | $94.66 \pm 0.75$ | $93.08 \pm 0.94$ | $90.93 \pm 0.76$ | $89.95 \pm 1.31$ | $86.11 \pm 1.43$ | $86.95 \pm 1.46$ |
| LRI-Gaussian | $97.51 \pm 0.76$ | $95.69 \pm 0.80$ | $93.64 \pm 1.39$ | $91.42 \pm 2.97$ | $89.89 \pm 1.15$ | $87.45 \pm 1.47$ | $88.13 \pm 0.63$ |

GradGAM, BernMask-P and PointMask are unstable and can only provide some interpretability for one or two backbones, while GradGeo and BernMask seem to fail to perform well on SynMol.

## 5.4 PLBind: Protein-ligand Binding Affinity Prediction

This task is to predict whether a protein-ligand pair is of affinity $K_D < 10$ nM. Each point in a protein is an amino acid associated with a 3D coordinate, a categorical feature indicating the amino acid type, and two scalar features. Each point in a ligand is an atom associated with a 3D coordinate, a categorical feature indicating the atom type, and a scalar feature. Different from other datasets, each sample in PLBind contains two sets of points. So, for each sample, two encoders will be used to encode the ligand and the protein separately, and the obtained two embeddings will be added to make a prediction. As shown in Table 2, LRI-Bernoulli outperforms all other methods, while LRI-Gaussian achieves comparable performance on EGNN. This might indicate that to make good predictions on PLBind, the existence of certain groups of amino acids is more important than their exact locations. Interestingly, all other methods do not seem to perform well on PLBind. Moreover, all methods have low ROC AUC, which suggests only a part of but not the entire binding site is important to decide the binding affinity.

## 5.5 Generalization Performance and Ablation Studies of LRI

LRI-induced models can generalize better while being interpretable. As shown in Table 4, both LRI-induced models never degrade prediction performance and sometimes may even boost it compared with models trained without LRI, i.e., using empirical risk minimization (ERM). Moreover, LRI-induced models are more robust to distribution shifts as LRI is grounded on the IB principle. Table 5 shows a study with shifts on the numbers of particle tracks, where all models are trained on samples with 10 particle tracks, and tested on samples with a different number of (from 10 to 70) tracks. We observe LRI-induced models work consistently better than models trained naively. We also conduct ablation studies on the differentiable graph reconstruction module proposed specifically for geometric data in LRI-Gaussian, we find that without this module the interpretation performance of LRI-Gaussian may be reduced up to 49% on ActsTrack and up to 23% on SynMol, which shows the significance of this module. More details of the ablation study can be found in Appendix F.1.

## 6 Conclusion

This work systematically studies interpretable GDL models by proposing a framework *Learnable Randomness Injection* (LRI) and four datasets with ground-truth interpretation labels from real-world scientific applications. We have studied interpretability in GDL from the perspectives of *existence importance* and *location importance* of points, and instantiated LRI with LRI-Bernoulli and LRI-Gaussian to test the two types of importance, respectively. We observe LRI-induced models provide interpretation best aligning with scientific facts, especially LRI-Gaussian that tests location importance. Grounded on the IB principle, LRI never degrades model prediction performance, and may often improve it when there exist distribution shifts between the training and test scenarios.

ACKNOWLEDGMENTS

We greatly thank the actionable suggestions given by reviewers. S. Miao and M. Liu are supported by the National Science Foundation (NSF) award OAC-2117997. Y. Luo is partially supported by a 2022 Seed Grant Program of the Molecule Maker Lab Institute, an NSF AI Institute (grant no. 2019897) at the University of Illinois Urbana-Champaign. P. Li is supported by the JPMorgan Faculty Award.

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

## A  VARIATIONAL BOUNDS OF THE IB PRINCIPLE

Let's assume the sample with its label $(\mathcal{C}, Y) \sim \mathbb{P}_{\mathbb{C} \times \mathbb{Y}}$. We ignore the $\tilde{\mathcal{G}}$ in our objectives to keep the notation simple, then, the IB objective is:

$$\min -I(\tilde{\mathcal{C}}; Y) + \beta I(\tilde{\mathcal{C}}; \mathcal{C}), \tag{3}$$

where $\tilde{\mathcal{C}}$ is the perturbed sample, and $I(.;.)$ denotes the mutual information between two random variables.

For the first term $-I(\tilde{\mathcal{C}}; Y)$, by definition we have:

$$-I(\tilde{\mathcal{C}}; Y) = -\mathbb{E}_{\tilde{\mathcal{C}},Y} \left[ \log \frac{\mathbb{P}(Y \mid \tilde{\mathcal{C}})}{\mathbb{P}(Y)} \right]. \tag{4}$$

We introduce a variational approximation $\mathbb{P}_\theta(Y \mid \tilde{\mathcal{C}})$ for $\mathbb{P}(Y \mid \tilde{\mathcal{C}})$ as it is intractable. Then, we yield a variational upper bound:

$$
\begin{aligned}
-I(\tilde{\mathcal{C}}; Y) &= -\mathbb{E}_{\tilde{\mathcal{C}},Y} \left[ \log \frac{\mathbb{P}_\theta(Y \mid \tilde{\mathcal{C}})}{\mathbb{P}(Y)} \right] - \mathbb{E}_{\tilde{\mathcal{C}}} \left[ D_{\mathrm{KL}}(\mathbb{P}(Y \mid \tilde{\mathcal{C}}) \| \mathbb{P}_\theta(Y \mid \tilde{\mathcal{C}})) \right] \\
&\leq -\mathbb{E}_{\tilde{\mathcal{C}},Y} \left[ \log \frac{\mathbb{P}_\theta(Y \mid \tilde{\mathcal{C}})}{\mathbb{P}(Y)} \right] \\
&= -\mathbb{E}_{\tilde{\mathcal{C}},Y} \left[ \log \mathbb{P}_\theta(Y \mid \tilde{\mathcal{C}}) \right] - H(Y),
\end{aligned}
\tag{5}
$$

where $H(Y)$ is the entropy of $Y$ which is a constant. We use the prediction model $f$ paired with the cross-entropy loss $\mathcal{L}_{CE}(f(\tilde{\mathcal{C}}), Y)$ to represent $-\mathbb{E}_{\tilde{\mathcal{C}},Y} \left[ \log \mathbb{P}_\theta(Y \mid \tilde{\mathcal{C}}) \right]$, minimizing which is thus equivalent to minimizing a variational upper bound of $-I(\tilde{\mathcal{C}}; Y)$.

For the second term $I(\tilde{\mathcal{C}}; \mathcal{C})$, because $\tilde{\mathcal{C}} = g(\mathcal{C})$. Suppose $\phi$ is the parameter of $g$. By definition, we have:

$$I(\tilde{\mathcal{C}}; \mathcal{C}) = \mathbb{E}_{\tilde{\mathcal{C}},\mathcal{C}} \left[ \log \frac{\mathbb{P}_\phi(\tilde{\mathcal{C}} \mid \mathcal{C})}{\mathbb{P}(\tilde{\mathcal{C}})} \right]. \tag{6}$$

As $\mathbb{P}(\tilde{\mathcal{C}})$ is intractable, we introduce a variational approximation $\mathbb{Q}(\tilde{\mathcal{C}})$. Then, we yield a variational upper bound:

$$
\begin{aligned}
I(\tilde{\mathcal{C}}; \mathcal{C}) &= \mathbb{E}_{\tilde{\mathcal{C}},\mathcal{C}} \left[ \log \frac{\mathbb{P}_\phi(\tilde{\mathcal{C}} \mid \mathcal{C})}{\mathbb{Q}(\tilde{\mathcal{C}})} \right] - D_{\mathrm{KL}} \left( \mathbb{P}(\tilde{\mathcal{C}}) \| \mathbb{Q}(\tilde{\mathcal{C}}) \right) \\
&\leq \mathbb{E}_{\mathcal{C}} \left[ D_{\mathrm{KL}} \left( \mathbb{P}_\phi(\tilde{\mathcal{C}} \mid \mathcal{C}) \| \mathbb{Q}(\tilde{\mathcal{C}}) \right) \right].
\end{aligned}
\tag{7}
$$

For LRI-Bernoulli, $g_\phi$ takes as input $\mathcal{C} = (\mathcal{V}, \mathbf{X}, \mathbf{r})$ and first outputs $p_v \in [0, 1]$ for each point $v \in \mathcal{V}$. Then, it samples $m_v \sim \mathrm{Bern}(p_v)$ and yields $\tilde{\mathcal{C}}$ by removing all points with $m_v = 0$ in $\mathcal{C}$. This procedure gives $\mathbb{P}_\phi(\tilde{\mathcal{C}} \mid \mathcal{C}) = \prod_{v \in \mathcal{V}} \mathbb{P}(m_v | p_v)$, which essentially makes $m_v$ conditionally independent across different points given the input point cloud $\mathcal{C}$. In this case, we define $\mathbb{Q}(\tilde{\mathcal{C}})$ as follows. For every point cloud $\mathcal{C} \sim \mathbb{P}_{\mathbb{C}}$, we sample $m'_v \sim \mathrm{Bern}(\alpha)$, where $\alpha \in [0, 1]$ is a hyperparameter. We remove all points in $\mathcal{C}$ and add points when their $m'_v = 1$. This procedure gives $\mathbb{Q}(\tilde{\mathcal{C}}) = \sum_{\mathcal{C}} \mathbb{P}(\mathbf{m}' \mid \mathcal{C}) \mathbb{P}(\mathcal{C})$. As $\mathbf{m}'$ is independent from $\mathcal{C}$ given its size $n$, $\mathbb{Q}(\tilde{\mathcal{C}}) = \sum_n \mathbb{P}(\mathbf{m}'|n) \mathbb{P}(\mathcal{C} = n) = \mathbb{P}(n) \prod_{v=1}^n \mathbb{P}(m'_v)$, where $\mathbb{P}(n)$ is a constant. Then, we yield:

$$D_{\mathrm{KL}} \left( \mathbb{P}_\phi(\tilde{\mathcal{C}} \mid \mathcal{C}) \| \mathbb{Q}(\tilde{\mathcal{C}}) \right) = \sum_{v \in \mathcal{V}} D_{\mathrm{KL}}(\mathrm{Bern}(p_v) \| \mathrm{Bern}(\alpha)) + c(n, \alpha), \tag{8}$$

where $c(n, \alpha)$ does not contain parameters to be optimized. Therefore, minimizing the second term of LRI-Bernoulli is equivalent to minimizing a variational upper bound of $I(\tilde{\mathcal{C}}; \mathcal{C})$. Now, we can conclude that the objective of LRI-Bernoulli is a variational upper bound of the IB principle.

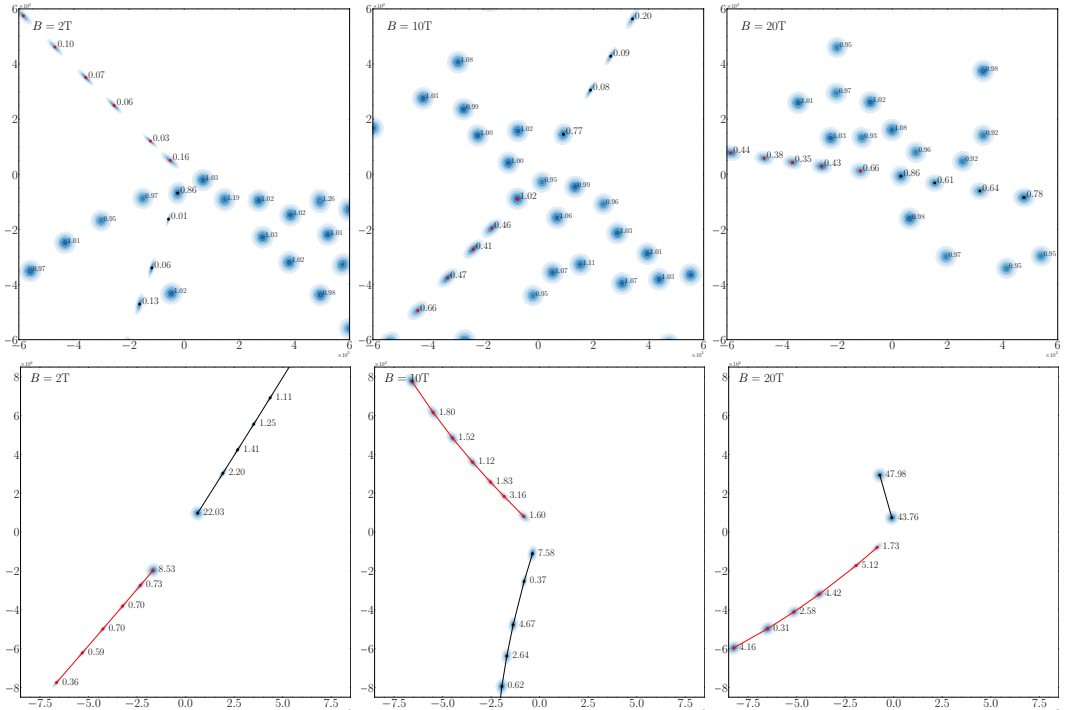

Figure 4: Visualizing interpretable patterns discovered by LRI-Gaussian for `ActsTrack`, where the origins $(0, 0)$ are the centers of the collisions. Points are plotted in x-y plane (mm) with magnetic fields perpendicular to the plane, only points colored black/red indicate the two tracks of the two $\mu$'s from the $z \rightarrow \mu\mu$ decay, and the blue contour superimposed on each point $v$ is the shape of the (scaled) learned Gaussian $\mathcal{N}(\mathbf{0}, \mathbf{\Sigma}'_v)$, where $\mathbf{\Sigma}'_{v_{i,j}} = 400\mathbf{\Sigma}_{v_{i,j}}$, i.e., each raw element is multiplied by $400$, as otherwise the contour would be hardly visible. For figures in the top row, the number next to each point $v$ is $|\mathbf{\Sigma}_v|$; for figures in the bottom row, the number next to each point is the angle between the true track direction and the learned direction, and non-critical points are not plotted for better visualization.

For LRI-Gaussian, $g_\phi$ now takes as input $\mathcal{C} = (\mathcal{V}, \mathbf{X}, \mathbf{r})$ and first outputs a covariance matrix $\mathbf{\Sigma}_v \in \mathbb{R}^{3 \times 3}$ for each point $v \in \mathcal{V}$. Then, it samples $\epsilon_v \sim \mathcal{N}(\mathbf{0}, \mathbf{\Sigma}_v)$ and yields $\tilde{\mathcal{C}} = (\mathcal{V}, \mathbf{X}, \mathbf{r} + \epsilon)$, where $\epsilon$ is a vector containing $\epsilon_v$ for each point. This procedure gives $\mathbb{P}_\phi(\tilde{\mathcal{C}} \mid \mathcal{C}) = \prod_{v \in \mathcal{V}} \mathbb{P}(\epsilon_v | \mathbf{\Sigma}_v)$, which essentially makes $\epsilon_v$ conditionally independent across different points given the input point cloud $\mathcal{C}$. In this case, we define $\mathbb{Q}(\tilde{\mathcal{C}})$ as follows. For every point cloud $\mathcal{C} \sim \mathbb{P}_\mathbb{C}$, we sample $\epsilon'_v \sim \mathcal{N}(\mathbf{0}, \sigma\mathbf{I})$ and yield $\tilde{\mathcal{C}} = (\mathcal{V}, \mathbf{X}, \mathbf{r} + \epsilon')$, where $\sigma$ is a hyperparameter. Similarly, we obtain $\mathbb{Q}(\tilde{\mathcal{C}}) = \mathbb{P}(n) \prod_{v=1}^{n} \mathbb{P}(\epsilon'_v)$, where $\mathbb{P}(n)$ is a constant. Finally, for this case, we have:

$$D_{\mathrm{KL}} \left( \mathbb{P}_\phi(\tilde{\mathcal{C}} \mid \mathcal{C}) \| \mathbb{Q}(\tilde{\mathcal{C}}) \right) = \sum_{v \in \mathcal{V}} D_{\mathrm{KL}}(\mathcal{N}(\mathbf{0}, \mathbf{\Sigma}_v) \| \mathcal{N}(\mathbf{0}, \sigma\mathbf{I})) + c(n, \alpha), \tag{9}$$

where $c(n, \alpha)$ is a constant, therefore, minimizing the second term of LRI-Gaussian is equivalent to minimizing a variational upper bound of $I(\tilde{\mathcal{C}}; \mathcal{C})$. Hence, the objective of LRI-Gaussian is also a variational upper bound of the IB principle.

## B  INTERPRETATION VISUALIZATION

**ActsTrack.** Figure 4 shows the interpretable patterns discovered by LRI-Gaussian. The top row shows that the randomness of the learned Gaussian noise $\mathcal{N}(\mathbf{0}, \mathbf{\Sigma}_v)$ indicates the importance of each point $v$, and thus $|\mathbf{\Sigma}_v|$ can measure the importance quantitatively as it characterizes the entropy of Gaussian. In the examples, non-critical points learn isotropic Gaussian close to $\mathcal{N}(\mathbf{0}, \sigma\mathbf{I})$ as regularized by the KL divergence in Eq. 2, while critical points left by the $\mu$'s produced by the $z \rightarrow \mu\mu$ decay learn multivariate Gaussian with small randomness, i.e., small $|\mathbf{\Sigma}_v|$, because high

randomness on these points would greatly affect model prediction and the prediction loss in Eq. 2 would not allow this to happen. As shown quantitatively in Table 2, $|\boldsymbol{\Sigma}_v|$ measures the importance of each point very well and can provide the best interpretation AUC in most cases.

The bottom row of Figure 4 shows that LRI-Gaussian is also capable of discovering fine-grained geometric patterns. In the examples, a small perturbation on each point along the underlying track direction may not even affect the model prediction (because the track radius may not change a lot by such perturbations). Therefore, the principal component (i.e., the eigenvector that corresponds to the largest eigenvalue) of $\boldsymbol{\Sigma}_v$ in x-y space can give an estimation of the track direction at $v$. As shown quantitatively in Table 3, the principal component can actually estimate the track direction very well. Meanwhile, we can also see that the ratio between the lengths of two principal components (i.e., $\frac{2^{\text{nd}} \text{ Largest Eigenvalue}}{\text{Largest Eigenvalue}}$) of $\boldsymbol{\Sigma}_v$ varies when the strength of the background magnetic field $B$ changes, and as shown quantitatively in Figure 3, this ratio provides an almost accurate estimation of $B$ up to a constant multiplier. This is because the direction orthogonal to the track direction is the direction of Lorentz force, and when the principal component of $\boldsymbol{\Sigma}_v$ in x-y space indicates the track direction, the second principal component in x-y space would be the direction of Lorentz force. Since Lorentz force $F \propto B$ and the properties of $\mu$'s (e.g., momentum) from the $z \rightarrow \mu\mu$ decay is independent of magnetic field strength, the ratio $\frac{2^{\text{nd}} \text{ Largest Eigenvalue}}{\text{Largest Eigenvalue}}$ can imply the strength of the background magnetic field.

## C  DETAILS OF THE DATASETS

### C.1  BACKGROUND AND TASKS

We describe the background and tasks of our datasets in more detail in this section.

**ActsTrack.** As illustrated in Fig. 1a, protons will collide at the center of the detectors, and a number of interactions will happen, where different interactions may produce different particles, e.g., a $z \rightarrow \mu\mu$ decay produces two $\mu$'s. The produced particles will then fly through multiple layers of detectors with a magnetic field, and Fig. 1a shows an example that has four layers of ring detectors with a 2T magnetic field parallel to z-axis. When a particle flies through such detectors, the detectors will record where the particle hits the detector (i.e., geometric coordinates) and measure certain properties of the particle (e.g., momentum) depending on the type of the detector. And if the particle is charged, then its track in space may be curved due to the magnetic field. As such, each particle will leave a set of points on the detectors with some measured properties, and all particles produced from the collision will then form a point cloud, where a point is just a particle hit on a detector. However, we are only interested in a certain interaction, i.e., the $z \rightarrow \mu\mu$ decay, and do not care other interactions (these interactions are called pileup interactions or background events). Fortunately, the particle hits in the point cloud will reveal the information about the interactions happened in the collision. For example, if a $z \rightarrow \mu\mu$ decay just happened, then there should be two sets of points in the point cloud that are left by the two $\mu$'s just produced from the decay. In principle, it is possible to reconstruct the type of a particle based on its tracks and/or some other measured properties in a magnetic field, and once we can find that there are two tracks that must be caused by two $\mu$'s (with certain invariant mass), then we would know a $z \rightarrow \mu\mu$ decay just happened in the collision. Therefore, the *classification* task of `ActsTrack` is to predict if there exists a $z \rightarrow \mu\mu$ decay or not in the collision given the point cloud measured by the detectors. Each positive sample contains particle hits from both a $z \rightarrow \mu\mu$ decay and some pileup interactions, while each negative sample has only hits from pileup interactions. In principle, if a classifier can successfully predict the existence of a $z \rightarrow \mu\mu$ decay, it should be aware of which sets of points in the point cloud represent the tracks of the $\mu$'s. Therefore, the particle hits left by the two $\mu$'s are labeled as ground-truth interpretation, and we expect an interpretable model to highlight these points as important points for the classification task.

**Tau3Mu.** Even though `Tau3Mu` has a similar basic HEP setup as `ActsTrack`, as shown in Fig. 1b, `Tau3Mu` now has a different configuration of the detectors and the magnetic field. Besides, the underlying physics process of interest is different and the task is with a different physics motivation (as shown in Sec. 1). The formulation of the machine learning task is a *classification* task to predict if there exists a $\tau \rightarrow \mu\mu\mu$ decay or not given a point cloud. Each positive sample contains both a $\tau \rightarrow \mu\mu\mu$ decay and some pileup interactions, while each negative sample has only pileup interactions.

Likewise, if a classifier can successfully predict whether there is a $\tau \to \mu\mu\mu$ decay or not, we expect those particle hits left by the three $\mu$'s produced by the $\tau \to \mu\mu\mu$ decay to be the important points for the classification task, and an interpretable model should be able to highlight these points.

**SynMol.** Different from typical molecular property prediction tasks, `SynMol` provides 3D representations of molecules instead of 2D chemical bond graphs. Namely, each molecule is represented as a 3D point cloud with each point being an atom in space. The *classification* task of `SynMol` is to predict if a molecule has a certain property or not. As shown in Fig. 1c, we know that the property of interest is determined by two certain functional groups (i.e., specific groupings of atoms). Therefore, we expect an interpretable model that can successfully predict the property should be able to highlight those atoms in the two functional groups as the important points for the classification task.

**PLBind.** The *classification* task of `PLBind` is to predict if a pair of protein and ligand can bind together (with a high affinity) or not given the 3D structures of proteins and ligands. Notably, Fig. 1d only shows the surface of a protein for better visualization, and actually proteins in `PLBind` are represented as 3D point clouds, where each point is just an amino acid in the protein. Similarly, ligands are just small molecules, and they are also represented as 3D point clouds, where each point is an atom in the ligand. In principle, as shown in Fig. 1d, a high binding affinity should largely depend on the amino acids of the protein that are near the binding site. Therefore, we label those amino acids near the binding site as ground-truth interpretation, and we expect an interpretable classifier that can successfully predict the high-affinity pairs should be able to highlight those amino acids as important points for the classification task.

## C.2 DATASET COLLECTION

We elaborate how we collect the four datasets in this section. The statistics of the four datasets are shown in Table 1.

**Basic Settings.** 1) For each sample in the four datasets, all points are centered at the origin. 2) Only positive samples in our datasets have ground-truth interpretation labels, so we only evaluate interpretation performance on positive samples. 3) For any pair of points $v$ and $u$ in the four datasets, they have an edge feature of $\left(\|\mathbf{r}_v - \mathbf{r}_u\|, \frac{\mathbf{r}_v - \mathbf{r}_u}{\|\mathbf{r}_v - \mathbf{r}_u\|}\right)$ if they are connected in the constructed $k$-nn graph.

**ActsTrack.** $z \to \mu\mu$ events are simulated with PYTHIA generator (Bierlich et al., 2022) overlaid with soft QCD pileup events, and particle tracks are simulated using Acts Common Tracking Software (Ai et al., 2022). For all samples, ten pileup interactions are generated with a center-of-mass energy of 14TeV, and the additional hard scatter interaction is only generated for positive samples. Particle tracks are simulated with a magnetic field parallel to the z axis, and the default `ActsTrack` is simulated with $B = 2$T . To make sure both $\mu$'s from the decay are properly measured, we calculate the invariant mass $m_{ij}$ for every pair of $\mu$'s in the generated data using Eq. 10 so that every positive sample in our dataset has at least a pair of $\mu$' with an invariant mass close to 91.19 GeV, i.e., the mass of the $z$ bosons.

$$\frac{1}{2}m_{ij}^2 = m^2 + \left(\sqrt{m^2 + p_{x,i}^2 + p_{y,i}^2 + p_{z,i}^2} \cdot \sqrt{m^2 + p_{x,j}^2 + p_{y,j}^2 + p_{z,j}^2}\right)$$
$$- \left(p_{x,i} \cdot p_{x,j} + p_{y,i} \cdot p_{y,j} + p_{z,i} \cdot p_{z,j}\right). \tag{10}$$

Then, those detector hits left by the $\mu$'s from the $z \to \mu\mu$ decay is labeled as ground-truth interpretation. As our focus is on model interpretation performance, we only keep 10 tracks in each sample to train and test models to reduce classification difficulty, while raw files with all tracks are also provided. In the future works, we will study a more extensive evaluation of different approaches over the samples with all tracks. Each point in a sample is a detector hit left by some particle, and it is associated with a 3D coordinate. As points in `ActsTrack` do not have any features in $\mathbf{X}$, we use a dummy $\mathbf{X}$ with all ones when training models. Momenta of particles measured by the detectors are also provided, which can help evaluate fine-grained geometric patterns in the data, but it is not used for model training. Because each particle on average leaves 12 hits and a model may classify samples well if it captures the track of any one of the $\mu$'s from the decay, we report both precision@12 and precision@24. Finally, we randomly split the dataset into training/validation/test sets with a ratio of $70 : 15 : 15$.

**Tau3Mu.** The $\tau$ leptons produced in decays of D and B mesons simulated by the PYTHIA generator (Bierlich et al., 2022) are used to generate the signal samples. The background events are generated with multiple soft QCD interactions modeled by the PYTHIA generator with a setting that resembles the collider environment at the High Luminosity LHC. The generated muons' interactions with the material in the endcap muon chambers are simulated including multiple scattering effects that resemble the CMS detector. The signal sample is mixed with the background samples at per-event level (per point cloud), with the ground-truth labels preserved for the interpretation studies. The hits left by the $\mu$'s from the $\tau \rightarrow \mu\mu\mu$ decay are labeled as ground-truth interpretation. We only use hits on the first layer of detectors to train models and make sure every sample in the dataset has at least three detector hits. Each point in the sample contains measurements of a local bending angle and a 2D coordinate in the pseudorapidity-azimuth ($\eta$-$\phi$) space. Because in the best case the model only needs to capture hits from each $\mu$, we report precision@3. And because $80\%$ of positive samples have less than 7 hits labeled as ground-truth interpretation, we also report precision@7. Finally, we randomly split the dataset into training/validation/test sets with a ratio of $70 : 15 : 15$.

**SynMol.** We utilize the molecules in ZINC (Irwin et al., 2012), and follow McCloskey et al. (2019) and Sanchez-Lengeling et al. (2020) to create synthetic properties based on the existence of certain functional groups. Specifically, if a molecule contains both the unbranched alkane and carbonyl, then we label it as a positive sample; otherwise it is labeled as a negative sample. So, the atoms in branched alkanes and carbonyl are labeled as ground-truth interpretation. Instead of 2D representations of molecules, we associate each atom a 3D coordinate by generating a conformer for each molecule. To do this, we first add hydrogens to the molecule and apply the ETKDG method (Riniker & Landrum, 2015). After that, the generated structures are cleaned up using the MMFF94 force field (Halgren, 1999) with a maximum iteration of 1000, and the added hydrogens are removed once it is finished. Both ETKDG and MMFF94 are implemented using RDKit. Besides a 3D coordinate, each point in a sample also has a categorical feature indicating the atom type. Even though the two functional groups may only have five atoms in total, some molecules may contain multiple such functional groups. So, we report both precision@5 and precision@8 ($80\%$ of positive samples have less than 8 atoms labeled as ground-truth interpretation). Finally, we split the dataset into training/validation/test sets in a way that the number of molecules with or without either of these functional groups is uniformly distributed following (McCloskey et al., 2019) so that the dataset bias is minimized.

**PLBind.** We utilize protein-ligand complexes from PDBBind (Liu et al., 2017), which annotates binding affinities for a subset of complexes in the Protein Data Bank (PDB) (Berman et al., 2000). In PDBBind, each protein-ligand pair is annotated with a dissociation constant $K_d$, which measures the binding affinity between a pair of protein and ligand. We use a threshold of 10 nM on $K_d$ to obtain a binary classification task, and the model interpretability is studied on the protein-ligand pairs with high affinities. To augment negative data, during training, there is a $10\%$ change of switching the ligand of a complex to a random ligand, and the new protein-ligand pair will be labeled as a negative sample, i.e., low affinity. The ground-truth interpretation labels consist of two parts. First, as shown in previous studies (Liu et al., 2022), using the part of the protein that is within 15Å of the ligand is enough to even learn to generate ligands that bind to a certain protein, so, we define the amino acids that are within 15Å of the ligand to be the binding site and label them as ground-truth interpretation. Second, we retrieve all atomic contacts (hydrogen bond and hydrophobic contact) for every protein-ligand pair from PDBsum (Laskowski et al., 2018) and label the corresponding amino acids in the protein as the ground-truth interpretation. Each amino acid in a protein is associated with a 3D coordinate, the amono acid type, solvent accessible surface area (SASA), and the B-factor. Each atom in a ligand is associated with a 3D coordinate, the atom type, and Gasteiger charges. Finally, we split the dataset into training/validation/test sets according to the year the complexes are discovered following Stärk et al. (2022).

## D DETAILS ON HYPERPARAMETER TUNING

All hyperparameters are tuned based on validation classification AUC for a fair comparison. All settings are trained with 5 different seeds and the average performance on the 5 seeds are reported.

**Basic Settings.** We use a batch size of 128 on all datasets, except on `Tau3Mu` we use a batch size of 256 due to its large dataset size. The Adam (Kingma & Ba, 2015) optimizer with a learning rate

Table 6: Ablation studies on the effect of the differentiable graph reconstruction module in LRI-Gaussian on `ActsTrack` and `SynMol`.

| `ActsTrack` | DGCNN | | | Point Transformer | | | EGNN | | |
|---|---|---|---|---|---|---|---|---|---|
| | ROC AUC | Prec@12 | Prec@24 | ROC AUC | Prec@12 | Prec@24 | ROC AUC | Prec@12 | Prec@24 |
| w/ Graph Recons. | $94.74 \pm 0.48$ | $91.01 \pm 1.02$ | $78.47 \pm 0.83$ | $91.89 \pm 2.07$ | $88.05 \pm 2.74$ | $74.09 \pm 3.61$ | $94.43 \pm 0.75$ | $90.85 \pm 0.69$ | $76.42 \pm 0.80$ |
| w/o Graph Recons. | $67.03 \pm 1.82$ | $56.09 \pm 2.11$ | $41.52 \pm 0.79$ | $60.64 \pm 1.93$ | $39.16 \pm 2.67$ | $32.34 \pm 1.42$ | $82.50 \pm 0.54$ | $78.71 \pm 3.84$ | $59.02 \pm 1.37$ |

| `SynMol` | DGCNN | | | Point Transformer | | | EGNN | | |
|---|---|---|---|---|---|---|---|---|---|
| | ROC AUC | Prec@5 | Prec@8 | ROC AUC | Prec@5 | Prec@8 | ROC AUC | Prec@5 | Prec@8 |
| w/ Graph Recons. | $98.83 \pm 0.63$ | $95.91 \pm 1.73$ | $76.86 \pm 0.62$ | $95.81 \pm 1.42$ | $89.61 \pm 3.30$ | $72.84 \pm 2.12$ | $96.88 \pm 1.25$ | $89.17 \pm 5.03$ | $74.47 \pm 1.05$ |
| w/o Graph Recons. | $98.65 \pm 0.39$ | $95.98 \pm 1.43$ | $76.58 \pm 0.33$ | $90.06 \pm 1.51$ | $79.80 \pm 0.80$ | $64.25 \pm 1.31$ | $78.39 \pm 36.27$ | $66.26 \pm 35.52$ | $58.79 \pm 30.07$ |

of $1.0 \times 10^{-3}$ and a weight decay of $1.0 \times 10^{-5}$ are used. `ActsTrack`, `SynMol`, and `PLBind` construct $k$-nn graphs with $k$ being 5; `Tau3Mu` constructs the graph by drawing edges for any pair of points with a distance less than 1. For a fair comparison, all models will be trained with 300 epochs on `ActsTrack` and `SynMol` and will be trained with 100 epochs on `Tau3Mu` and `PLBind`, so that all models are converged. For post-hoc methods, a classifier will be first pretrained with the epochs mentioned above, and those methods will further work on the model from the epoch with the best validation classification AUC during pretraining.

**LRI-Bernoulli.** $\beta$ is tuned from $\{1.0, 0.1, 0.01\}$ after normalizing the summation of the KL divergence by the total number of points. $\alpha$ is tuned from $\{0.5, 0.7\}$. Note that $\alpha$ should not be less than 0.5, because $\text{Bern}(0.5)$ injects the most randomness from an information-theoretic perspective. For `ActsTrack` and `SynMol`, we either train LRI-Bernoulli 300 epochs from scratch or first pre-train $f$ with 200 epochs and then train both $f$ and $g$ with 100 epochs (also based on the best validation prediction performance). Similarly, for `Tau3Mu` and `PLBind`, we either train it 100 epochs from scratch or first pre-train $f$ by 50 epochs and then train both $f$ and $g$ by 50 epochs. The temperature in the Gumbel-softmax trick is not tuned and is set to 1.

**LRI-Gaussian.** $\beta$ and the training time is tuned in the same way as for LRI-Bernoulli. Instead of directly tuning $\sigma$, we tune it by rescaling the magnitude of the geometric features to make sure the KL divergence is stable for optimization, because the magnitude of $\mathbf{r}$ may vary significantly across different datasets, e.g., extremely small or large. Specifically, we keep $\sigma = 1$ and rescale $\mathbf{r}$ by multiplying a constant $c$ on it so that 1 would be roughly the $5^{\text{th}}$ or $10^{\text{th}}$ percentile of the entries in $\text{abs}(c \cdot \mathbf{r})$. In this way, $c$ is tuned from $\{200, 300\}$ for `ActsTrack`, $\{10, 15\}$ for `SynMol`, $\{7, 11\}$ for `Tau3Mu`, and $\{0.9, 1.2\}$ for `PLBind`.

**BernMask.** This approach generalizes GNNExplainer (Ying et al., 2019) and learns a node mask $\mathbf{m} \in [0, 1]^n$ for each sample $\mathcal{C}$. Similar to GNNExplainer, there are two regularizers in its loss function, that is, mask size and entropy constraints. Specifically, the coefficient of the $\ell_1$ penalty on the normalized mask size, i.e., $\frac{\|\mathbf{m}\|_1}{n}$, is tuned from $\{1.0, 0.1, 0.01\}$. Element-wise entropy of $\mathbf{m}$ is applied, i.e., $\frac{1}{n} \sum_i -m_i \log m_i - (1 - m_i) \log(1 - m_i)$ and $m_i$ is the $i^{th}$ entry of $\mathbf{m}$, to encourage generating discrete masks, and the coefficient of this entropy constraint is tuned from $\{1.0, 0.1, 0.01\}$. After pretraining the classifier, we set the number of iterations to learn masks per sample as 500 and tune the learning rate for each iteration from $\{0.1, 0.001\}$.

**BernMask-P.** This approach generalizes PGExplainer (Luo et al., 2020) and learns a node mask $\mathbf{m} \in [0, 1]^n$ for each sample $\mathcal{C}$. Similar to BernMask, it also has mask size and entropy constraints in its loss function, and their coefficients are both tuned from $\{1.0, 0.1, 0.01\}$ as well. After pretraining the classifier, the explainer module in BernMask-P will be trained with extra 100 epochs on `ActsTrack` and `SynMol` and with extra 50 epochs on `Tau3Mu` and `PLBind`. The temperature in the Gumbel-softmax trick is set to 1.

**PointMask** (Taghanaki et al., 2020)**.** It is trained with 300 epochs on `ActsTrack` and `SynMol`, and with 100 epochs on `Tau3Mu` and `PLBind`. It has two hyperparameters as specified in their paper, i.e., $\alpha$ and $t$, where $\alpha$ is tuned from $\{1.0, 0.1, 0.01\}$ and $t$ is tuned from $\{0.2, 0.5, 0.8\}$.

## E  IMPLEMENTATION DETAILS

**Backbone Models.** All backbone models have 4 layers with a hidden size of 64. Batch normalization (Ioffe & Szegedy, 2015) and ReLU activation (Nair & Hinton, 2010) are used. All MLP layers use a dropout (Srivastava et al., 2014) ratio of 0.2. SUM pooling is used to generate point

cloud embeddings to make predictions. All backbone models utilize implementations available in Pytorch-Geometric (PyG) (Fey & Lenssen, 2019).

**LRI-Bernoulli.** Directly removing points from $\mathcal{C}$ to yield $\tilde{\mathcal{C}}$ makes it non-differentiable. We provide two differentiable ways to approximate this step. The first way is to use another MLP to map the raw point features $\mathbf{X}$ to a latent feature space $\mathbf{H}$, and yield $\tilde{\mathcal{C}} = (\mathcal{V}, (\mathbf{m1}^T) \odot \mathbf{H}, \mathbf{r})$. Here $\mathbf{m}$ is a vector containing $m_v$ for each point $v \in \mathcal{V}$, $\mathbf{1}$ is a vector of all ones, and $\odot$ denotes element-wise product. This is because masking on $\mathbf{X}$ or $\mathbf{r}$ is inappropriate as values in them have specific physical meanings. When the backbone model is implemented by using a message passing scheme, e.g., using PyG, another possible way is to use $\mathbf{m}$ to mask the message sent by the points to be removed. We find both ways can work well and we adopt the second way in our experiments.

**LRI-Gaussian.** We use a softplus layer to output $a_1$ and $a_2$ to parameterize the Gaussian distribution. To make it numerically stable we clip the results of the softplus layer to $[1.0 \times 10^{-6}, 1.0 \times 10^{6}]$. For $\phi$ in the differentiable graph reconstruction module, we find it empirically a simple MLP can work well enough and thus we adopt it to implement $\phi$.

**Baseline Methods.** BernMask is extended from Ying et al. (2019) based the authors' code and the implementation available in PyG. BernMask-P is extended from Luo et al. (2020) based on the authors' code and a recent PR in PyG. PointMask is reproduced based on the authors' code. GradGAM and GradGeo are extended based on the code from Gildenblat & contributors (2021).

**Discussions on LRI.** We note that both $f$ and $g$ in LRI needs a permutation equivariant encoder to learn point representations, and we find for simple tasks these two encoders can share parameters to reduce model size without degrading interpretation performance, while for challenging tasks using two different encoders may be beneficial. In our experiments we use the same encoder for `ActsTrack`, `Tau3Mu`, and `SynMol`, and use two different encoders for `PLBind`. If the model size is not a concern, using two encoders can be a good starting point. The other thing is that one of the key components in LRI is the perturbation function $h$, as shown in Fig. 2, and we have shown two ways to design $h$, i.e., Bernoulli perturbation and Gaussian perturbation. Nonetheless, $h$ can be generalized in many different ways. For example, $h$ is where one can incorporate domain knowledge and provide contextualized interpretation results, i.e., human understandable results. For instance, instead of perturbing molecules in the atom level, it is possible to perturb molecules in the functional group level, e.g., by averaging the learned perturbation in each functional group, so that the interpretation results can be more contextualized and more human understandable. In this work, we experiment this feature on `PLBind` by replacing the learned $\{p_u\}_{u \in \mathcal{N}(v) \bigcup v}$ or $\{\mathbf{\Sigma}_u\}_{u \in \mathcal{N}(v) \bigcup v}$ in a neighbourhood with the minimum $p$ or with the $\mathbf{\Sigma}$ having the maximum determinant in that neighbourhood, which encourages either aggressively perturbing all amino acids in a neighborhood or not perturbing any amino acids in the neighborhood, and we find this can better help discover binding sites for `PLBind`.

## F  SUPPLEMENTARY EXPERIMENTS

### F.1  ABLATION STUDIES

Table 6 shows the effect of the differentiable graph reconstruction module in LRI-Gaussian, where $c$ is set to 200 on `ActsTrack` and to 5 on `SynMol` so that large perturbations are encouraged to better show the significance of this module, $\beta$ is set to 0.01, and $f$ is first trained by 200 epochs and then $f$ and $g$ are trained together by 100 epochs. We observe significant drops when LRI-Gaussian is trained without differentiable graph reconstruction, which matches our expectations and validates the necessity of the proposed module.

### F.2  FINE-GRAINED GEOMETRIC PATTERN LEARNING

To discover fine-grained geometric information in `ActsTrack` using LRI-Gaussian as shown in Table 3, we conduct experiments based on DGCNN, where $f$ is first trained with 100 epochs, and then $g$ is further trained with 500 epochs while $f$ is finetuned with a learning rate of $1e\text{-}8$. $f$ and $g$ use two different encoders, $\beta$ is set to 10, $c$ is 100, all dropout ratio is set to 0, and $\frac{\mathbf{r}_v}{\|\mathbf{r}_v\|}$ is used as

point features. Finally, we let $g$ directly output covariance matrix $\mathbf{\Sigma}_v \in \mathbb{R}^{2\times 2}$, i.e., in x-y space, to only perturb the first two dimensions of $\mathbf{r}$.

## G    LIMITATIONS AND FUTURE DIRECTIONS OF LRI

- Current LRI provides interpretation results at point-level, while cannot find out geometric patterns shared by a group of points (e.g., some set of points may be allowed to rotate around a reference point in space).

- Although Gaussian noise can help capture some fine-grained geometric patterns, if the underlying geometric patterns get too complicated, LRI-Gaussian may degrade to only offer a ranking of point importance and have limited capability to capture meaningful fine-grained geometric patterns. It may be possible to design customized noise with prior knowledge to detect different types of geometric patterns (e.g., what if some points can be freely moved along a curve without affecting prediction loss?), and can we even not use a fixed noise distribution (i.e., Gaussian or Bernoulli) and also make it trainable? Meanwhile, current LRI would need specific tuning to yield fine-grained interpretation results, which may be improved with better training strategies.

- Current LRI may provide interpretation results that are not contextualized enough to guide scientists for further research. For example, chemists may prefer an interpretable model to tell them directly which functional groups are important instead of which atoms are important.

