# OpenReview forum: "Interpretable Geometric Deep Learning via Learnable Randomness Injection"
_ICLR.cc/2023/Conference — ICLR 2023 poster_

### Official Review · Reviewer_HwR5 · 2022-10-21

**Confidence:** 3
**Correctness:** 3
**Technical Novelty And Significance:** 3
**Empirical Novelty And Significance:** 3
**Recommendation:** 8

**Clarity, Quality, Novelty And Reproducibility:**

I find the method clear, while some part of the specific tasks and the reported outcome seems not entirely accessible for non-expert readers. I consider both Novelty and Reproducibility positively. The authors attached the code, but I do not see statements about code and data release.

**Strength And Weaknesses:**

STRENGHT
=========
S1) NOVELTY AND RELEVANCE: While similar principles have been proposed in the past to study neural network interpretability (on graphs: Miao et al. 2022; on images: "U-noise: Learnable Noise Masks For Interpretable Image Segmentation", Koker et al., 2021), I am not aware of similar approaches in the field of 3D point clouds. The formulation looks straightforward and applicable also in contexts outside the datasets proposed in the paper. Improving interpretability in Geometric Deep learning is an active research field, and I believe the proposed methods might have a relevant impact.

S2) EXPERIMENTS: The two approaches are carefully tested with different backbones networks on several datasets, and also provide ablation and explanations of hyperparameter choices.

WEAKNESSES
========
W1) LIMITATIONS AND DISCUSSION: The paper does not offer an overview of methods limitations, nor discussion or visualization of the achieved interoperability for the different tasks. Some sentences are spread across the text, pointing to how to exploit the covariance matrix to obtain fine-grained geometric patterns, but without showing examples. Clarify the interpretability analysis and clearly stating the outcomes could strengthen the paper message.

W2) GEOMETRICAL DIVERSITY OF THE DATASETS: The proposed datasets present similar characteristics. The number of points for each sample is similar (between 15 and 340), with a similar proportion of importance points (between 20% and 30% of the entire point cloud) and the same number of classes (2 for all the datasets). It is also not clear if the different datasets face different points distribution (i.e., the density of the point clouds). I would suggest testing on data from different domains, coming from Computer Vision and Graphics (e.g., LiDAR, body scans, ShapeNet rigid objects), and also providing a measure about the density of the points w.r.t. the occupied space.


MINOR
=======
M1) Can LBI-Bernoulli and LBI-Gaussian be trained together? Does it introduce instability in training? I think that if no technical limitations exist, it would be an interesting validation.
M2) I think the methods can potentially work also on other domains (e.g., images, graphs), up to a minor reformulation of the two strategies. Is it correct? In that case, it might be mentioned as possible future work.

**Summary Of The Paper:**

The paper proposes two training schemas for fostering interpretability in Geometric Deep Learning by considering two properties: the relevance of a point presence and the relevance of a point location. A backbone classifier network is trained in an adversarial way where an Interpreter modifies the input data with a given amount of point-wise noise (either by removing points or perturbing their position). Then, the classifier module tries to improve the score by removing the noise while the Interpreter attempts to redistribute the amount of noise. In the end, the noise should be located at non-relevant points, while important points are kept unchanged. The method is tested across four different datasets from physic and chemistry, outperforming baselines in most cases.

**Summary Of The Review:**

While some aspects in the discussion and experiments of the results could be improved, I have a positive opinion of the work: the contribution is interesting, well-validated, and based on straightforward principles.

---

> ### Author Response · Authors · 2022-11-14
> **Response to Reviewer HwR5 (1/2)**
>
> We greatly thank Reviewer HwR5 for appreciating our contributions to interpretable GDL research, insightful suggestions on future studies, and strongly supporting the acceptance of this work. We address the raised questions as follows.
>
> > W1.1: The paper does not offer an overview of methods limitations.
>
> We thank the reviewer for raising this point. If the reviewer refers to the limitations of baseline methods, we address them [here](https://openreview.net/forum?id=6u7mf9s2A9&noteId=vEjJsn97zh1). In the following, we provide an overview of the potential limitations of our proposed LRI. We have also added our method limitations in Appendix F.
>
>
> - Current LRI provides interpretation results at point-level, while cannot find out geometric patterns shared by a group of points (e.g., some set of points may be allowed to rotate around a reference point in space).
> - Although Gaussian noise can help capture some fine-grained geometric patterns, if the underlying geometric patterns get too complicated, LRI-Gaussian may degrade to only offer a rank of point importance and have limited capability to capture meaningful fine-grained geometric patterns.
> It may be possible to design customized noise with prior knowledge to detect different types of geometric patterns (e.g., what if some points can be freely moved along a curve without affecting prediction loss?), and can we even not use a fixed noise distribution (i.e., Gaussian or Bernoulli) and also make it trainable?
> - Current LRI may provide interpretation results that are not contextualized enough to guide scientists for further research. For example, chemists may prefer an interpretable model to tell them directly which functional groups are important instead of which atoms are important.
>
>
> > W1.2: The paper does not offer a discussion or visualization of the achieved interpretability for the different tasks. Some sentences are spread across the text, pointing to how to exploit the covariance matrix to obtain fine-grained geometric patterns, but without showing examples. Clarify the interpretability analysis and clearly stating the outcomes could strengthen the paper message.
>
> We greatly thank the reviewer for the valuable suggestions to further strengthen the message of the paper. Due to limited time and the hardness of visualizing 3D data from scientific domains (with interpretation results), we promise that we will add examples, visualization, and discussion for all datasets, and we will at least add those for dataset ActsTrack (w/ examples of the discovered fine-grained patterns) before the end of the discussion period.
>
>
>
> > W2: The number of points for each sample is similar (between 15 and 340), with a similar proportion of importance points (between 20\% and 30\% of the entire point cloud) and the same number of classes (2 for all the datasets). It is also not clear if the different datasets face different points distribution (i.e., the density of the point clouds).
>
>
> We intentionally focus this work on scientific applications, because we believe model interpretability is more crucial for applications in scientific fields, where interpretable methods hold the promise to build more trustworthy models and further advance scientific discovery. Therefore, the characteristics of the datasets are determined by the nature of the corresponding scientific problems, but are not picked by us manually.
>
> Specifically, the number of points of each sample in our datasets is determined by the corresponding scientific domains, where the number of points in ActsTrack and Tau3mu is decided by the underlying physics processes, and the number of points in SynMol and PLBind is decided by the number of atoms/amino acids in the measured molecules/proteins.
> For example, ZINC [1], QM9 [4], and MoleculeNet [5] are widely used molecular property prediction datasets in scientific GDL, where, on average, each molecule has 23.2 atoms in ZINC, 18.0 atoms in QM9, and 25.5 atoms in MoleculeNet-HIV. Similarly, according to [7], the median protein length (number of amino acids) is 361 in eukarya, 267 in bacteria, and 247 in archaea organisms. Likewise, datasets from HEP may have 10-100 points with an average of 30–50 points in each sample for jet tagging tasks [8, 9], and each sample may have 6-12 points for event classification tasks [10, 11]. Therefore, our proposed datasets have similar sizes of point clouds to the typical applications in scientific GDL.

---

> > ### Author Response · Authors · 2022-11-14
> > **Response to Reviewer HwR5 (2/2)**
> >
> > Actually, our datasets have different proportions of important points from ~20\% to ~40\%,
> > but we agree that it would be beneficial to test interpretable GDL methods on point clouds with much more/fewer important points (e.g., 5\% or 80\%).
> >
> > In principle, our datasets should have quite different densities of points, because, for example, atoms/amino acids can be (relatively) close to each other in space due to atomic interactions, while particle hits from collisions in HEP can be located (relatively) far apart due to the momentum of particles. But we agree that it can be an interesting direction to study model interpretability in more detail when the densities of point clouds vary significantly, for which quantitative density measurements can be proposed/utilized and point clouds from Computer Vision and Graphics can be adopted.
> >
> > We agree that collecting datasets with more classes is a valuable future work to further facilitate interpretable GDL research.
> > As for the datasets proposed in this work, we formulate the tasks primarily based on their scientific motivations, and we did not intentionally constrain the number of classes. Binary classification just happens to be a quite effective formulation in scientific GDL, and this formulation can already serve the scientific purposes in many tasks [5, 6, 8].
> >
> >
> > We thank the reviewer for pointing out valuable future directions to enrich the datasets. As this is the first work that collects benchmark datasets for this domain, we can expect more people to join and make the benchmark datasets more comprehensive in the future.
> >
> >
> >
> > > M1: Can LBI-Bernoulli and LBI-Gaussian be trained together? Does it introduce instability in training?
> >
> > We have the same guess as the reviewer that it might cause instability in training if we naively train both LBI-Bernoulli and LBI-Gaussian together, and we agree that it can be an interesting direction to further explore the potential of the LRI framework. Currently, one simple way to make them work together is to utilize the idea of ensemble methods (e.g., weighted ranking).
> >
> >
> > > M2: The methods can potentially work also on other domains (e.g., images, graphs), up to a minor reformulation of the two strategies. Is it correct?
> >
> > Yes, we believe the idea of creating information bottlenecks and injecting learnable randomness may also benefit other domains. For example, it can be interesting to generalize LRI to tabular datasets, where both categorical features and continuous features may exist.
> >
> >
> > > M3: Some part of the specific tasks and the reported outcome seems not entirely accessible for non-expert readers.
> >
> > We thank the reviewer for pointing out this concern. Because of the page limitation, we address it by adding comprehensive descriptions of the background and tasks of our datasets in Appendix B, which should help readers to better understand the settings of the proposed datasets.
> >
> > > M4: The authors attached the code, but I do not see statements about code and data release.
> >
> > We have included the datasets in the folder `data` of the supplementary material. Currently, it contains three of the four datasets, and PLBind was not uploaded due to the limit of file size. We will release all codes and datasets in a Github repository later.
> >
> >
> > [1] Irwin, John J., et al. "ZINC: a free tool to discover chemistry for biology." Journal of chemical information and modeling. 2012.
> >
> > [2] Wang, Renxiao, et al. "The PDBbind database: Collection of binding affinities for protein-ligand complexes with known three-dimensional structures." Journal of medicinal chemistry. 2004.
> >
> > [3] Wu, Zhirong, et al. "3d shapenets: A deep representation for volumetric shapes." Proceedings of the IEEE conference on computer vision and pattern recognition. 2015.
> >
> > [4] Ramakrishnan, Raghunathan, et al. "Quantum chemistry structures and properties of 134 kilo molecules." Scientific data. 2014.
> >
> > [5] Wu, Zhenqin, et al. "MoleculeNet: a benchmark for molecular machine learning." Chemical science. 2018.
> >
> > [6] Townshend, Raphael JL, et al. "Atom3d: Tasks on molecules in three dimensions." Advances in Neural Information Processing Systems. 2021.
> >
> > [7] Brocchieri, Luciano, and Samuel Karlin. "Protein length in eukaryotic and prokaryotic proteomes." Nucleic acids research. 2005.
> >
> > [8] Qu, Huilin, and Loukas Gouskos. "Jet tagging via particle clouds." Physical Review D. 2020.
> >
> > [9] Qu, Huilin, Congqiao Li, and Sitian Qian. "Particle Transformer for Jet Tagging." International Conference on Machine Learning. 2022.
> >
> > [10] Ren, Jie, Lei Wu, and Jin Min Yang. "Unveiling CP property of top-Higgs coupling with graph neural networks at the LHC." Physics Letters B. 2020.
> >
> > [11] Abdughani, Murat, et al. "Probing the triple Higgs boson coupling with machine learning at the LHC." Physical Review D. 2021.

---

> > > ### Author Response · Authors · 2022-11-19
> > > **Updated manuscript provides examples for ActsTrack now**
> > >
> > > Dear Reviewer HwR5,
> > >
> > > We just included more descriptions and examples of LRI-Gaussian for dataset ActsTrack in Appendix G, and we will continue to work on providing more interpretation visualizations.
> > >
> > > Looking forward to your further comments.
> > >
> > > Best,
> > >
> > > Authors

---

> > > > ### Comment · Reviewer_HwR5 · 2022-11-23
> > > > **Post-Rebuttal**
> > > >
> > > > Thank the authors for their effort in replying to other reviewers and me.
> > > >
> > > > My initial worries were mainly on limitations discussion, the limited variety of the considered datasets, and some clarity aspects in the presentation. Other reviewers raised concerns about the presentation, while Reviewer Wziv also posed some methodological criticisms, keeping the score still towards acceptance.
> > > >
> > > > The rebuttal addresses the majority of the concerns: the only open points are still testing on different kinds of data, but I agree that the provided analysis is already satisfying, and it is reasonable to leave these kinds of studies for future works. The answer to Reviewer Wziv also clarifies the contribution of the paper - while I would like to know from the Reviewer if the doubts can be considered handled.
> > > >
> > > > Hence, I confirm my opinion and vote to accept the paper.

---

> > > > > ### Author Response · Authors · 2022-11-23
> > > > > **Thanks!**
> > > > >
> > > > > Thank you so much for your suggestions and appreciation, and we are also looking forward to other reviewers' responses.
> > > > >
> > > > > In the follow-up work, we will work on testing different kinds of data to further facilitate interpretable GDL research.

---

### Official Review · Reviewer_sW6V · 2022-10-23

**Confidence:** 4
**Correctness:** 4
**Technical Novelty And Significance:** 3
**Empirical Novelty And Significance:** 4
**Recommendation:** 8

**Clarity, Quality, Novelty And Reproducibility:**

The paper is well written and illustrated. The evaluation protocol is succinctly explained and the dataset gathering effort is appreciated by the reviewer. The work should be reproducible and easy to integrate.

Equation 2 can be simplified by penalizing deviation of the covariance from identity as the distributions are Gaussian?
Is the effect of \beta parameter discussed in the paper?

**Strength And Weaknesses:**

+ Important topic and a simple clean framework
+ End-to-end trainable wrt interpretability and classification
+ Benchmark datasets and good evaluation

The following are some of the weaknesses that can be explored in future works.
- Correlation across points is not considered. In many situation, this can be quite an important factor.
- Performance in adversarial setting is not discussed/explored.
- GradGAM, one of the gradient-based baselines, turns out to be a pretty good post-optimization approach.

**Summary Of The Paper:**

The submission develops two setups for building interpretable classifiers for geometric data and carefully applies them on four different test scenarios (from physics and medicine). The proposed models performs sensitivity analysis wrt either point inclusion (Bernoulli model) or point perturbation (Gaussian model). The proposed models are simple and can easily bee integrated with existing classifier models. The proposed models and classifier backbones are jointly trained. The paper also contributes benchmark datasets by collecting data from pretty diverse application settings and demonstrates the utility of the proposed framework.

I liked this submission as it demonstrates applications on non-standard dataset with potential implication beyond our immediate field.

**Summary Of The Review:**

Overall a good paper and I would be supportive of acceptance.

---

> ### Author Response · Authors · 2022-11-14
> **Response to Reviewer sW6V**
>
> We greatly thank Reviewer sW6V for appreciating our contributions to interpretable GDL research, insightful suggestions on future studies, and strongly supporting the acceptance of this work. We address the raised questions as follows.
>
> > W1: Correlation across points seems to be not considered.
>
> We are sorry for the confusion that one may have from our objectives Eq.(1) and Eq.(2). Actually, the added noise ($m_v$ in LRI-Bernoulli and $\epsilon_v$ in LRI-Gaussian) is conditionally independent across different points given the input point cloud $\mathcal{C}$, which indicates that the learned noise on different points will essentially be mutually dependent and thus can help capture the correlation across points. This is because when deriving our objectives, we assume the interpreter $g$ (parameterized by $\phi$) is $\mathbb{P}\_\phi(\tilde{\mathcal{C}} \mid \mathcal{C})=\prod_{v \in \mathcal{V}} \mathbb{P}\left(m_v \mid p_v\right)$ in LRI-Bernoulli and $\mathbb{P}\_\phi(\tilde{\mathcal{C}} \mid \mathcal{C})=\prod_{v \in \mathcal{V}} \mathbb{P}\left(\epsilon_v \mid \boldsymbol{\Sigma}_v\right)$ in LRI-Gaussian (see the paragraphs above Eq.(8) and Eq.(9) in Appendix A), which makes the noise to be added conditionally independent given $\mathcal{C}$. We have updated the Appendix to better present the message we obtained in the derivations of our objectives.
>
>
> > W2: Performance in adversarial setting is not discussed/explored.
>
> We totally agree that there can be more benefits brought by the information bottleneck principle as shown in previous studies, e.g., better robustness to adversarial attacks [1], and it would be an interesting future work to comprehensively test the robustness of LRI under adversarial settings (and distribution shifts).
>
> > W3: GradGAM, one of the gradient-based baselines, turns out to be a pretty good post-optimization approach.
>
> We are also surprised by this result, although LRI still largely outperforms it. We believe this is a very important observation, which indicates gradient-based baselines originally proposed for regular data (e.g., images) cannot be ignored even if the data of interest is irregular.
>
>
> > M1: Equation 2 can be simplified by penalizing deviation of the covariance from identity as the distributions are Gaussian?
>
> Yes, when implementing the objective, in practice one can use the simplified version. In the paper, we intend to present the general formulation of LRI using Eq.(1) and Eq.(2).
>
>
> > M2: Is the effect of $\beta$ parameter discussed in the paper?
>
> We find the performance is generally good when $\beta$ is in {$1, 0.1, 0.01$}, so we just view $\beta$ as a hyper-parameter that can be tuned from {$1, 0.1, 0.01$} based on the validation classification accuracy, and we have listed all tuned $\beta$ in the supplementary material (for both LRI-induced models and all three backbones),
> but we did not conduct further analysis on $\beta$.
>
> [1] Wu, Tailin, et al. "Graph information bottleneck." Advances in Neural Information Processing Systems. 2020.

---

### Official Review · Reviewer_Wziv · 2022-10-25

**Confidence:** 4
**Correctness:** 3
**Technical Novelty And Significance:** 3
**Empirical Novelty And Significance:** 3
**Recommendation:** 6

**Clarity, Quality, Novelty And Reproducibility:**

The paper is well-written, and it is a joy to read, especially the introduction. The proposed methods are clearly presented with sufficient details. Overall, it is a well-prepared submission with sound quality in clarity, quality, and reproducibility.

**Strength And Weaknesses:**

Strengths:
+ The paper is well-organized, written, and easy to follow.

+ The proposed research is well-motivated and appealing.

+ The proposed methods are simple and technically sound.

+ Four scientific datasets with interpretation ground truth will be released, which will benefit a large research community.

Weaknesses:
- Insufficient justification of technical contributions. Although the proposed methods, namely LRI-Bernoulli and LRI-Gaussian, are technically sound, it is hard to tell the technical contributions compared with existing works. For instance, PointMark, BernMask (GNNExplainer), GradGeo (DeepLIFT) are intensionally selected for comparison. How are they related to the proposed methods? What are the key differences/improvements from the proposed methods and previous mask-based methods (e.g., PointMark)?

- Computation overhead. Constructing KNN graphs is very time-consuming. The proposed LRI-Gaussian has to perform graph reconstruction after each round of perturbations. However, there is no discussion about computational cost either analytically or empirically.

- Missing ablation study. Without ablation studies, it is hard to tell whether the prediction and interpretation gains result from more parameters and additional computations.
The authors claim that existing interpretation methods cannot handle irregular geometric data. However, there is limited evidence to show that the proposed methods can handle it.
The authors claim that most existing methods can only provide either the existence or location importance. However, is this really a significant problem?
Can the proposed LRI-Bernoulli and LRI-Gaussian work together?

Minor Issues:
- The structures of g and f is unclear, how the f take points and 3D spatial coordinates to make predictions?
- How is the $| \Sigma_v|$ is used to rank the location importance?
- How the eigenvectors of $| \Sigma_v|$ is calculated?


**Summary Of The Paper:**

This paper proposes a framework for Learnable Randomness Injection (LRI) to train inherently interpretable geometric deep-learning models for scientific data. It perturbed data by learnable Bernoulli and Gaussian randomness masks in terms of existence and location importance. Furthermore, four scientific datasets with ground truth interpretation are proposed.

**Summary Of The Review:**

My major concern is the insufficient justification of technical contributions, computational cost analysis, and missing ablation study.

---

> ### Author Response · Authors · 2022-11-13
> **Response to Reviewer Wziv (1/4)**
>
> We greatly thank Reviewer Wziv for appreciating our contributions to interpretable GDL research, the insightful suggestions on further strengthening the manuscript, and supporting the acceptance of this work. We address the concerns as follows.
>
> > W1: Insufficient justification of technical contributions.
>
> We are sorry that we missed some technical details in the main text to demonstrate the technical contributions of our work, because we originally planned to let readers, especially non-ML domain experts, understand the methods without challenges.
>
> To the best of our knowledge, this is the first work that systematically studies interpretable GDL models. Specifically, LRI is the first general framework for building inherently interpretable GDL models that can well utilize geometric features and tackle data irregularity in GDL. LRI also enables testing both existence importance and location importance under the same framework.
> Moreover, LRI may even improve model generalization performance while providing excellent interpretability due to its connection to the IB principle, and is also the first work that can provide fine-grained interpretable geometric patterns. In addition, extending previous methods to the setting of GDL is also one of our technical contributions.
>
> Below we justify our technical contributions regarding LRI in more detail by reviewing the limitations of the baseline methods and comparing LRI with them.
>
> - GradCAM and GradGeo are based on two widely used gradient-based methods. These methods assume the sensitivity of a model to small continuous perturbations in features/feature maps can represent the importance of the corresponding features. Technically, this assumption is not justified in many scenarios, and empirically has been shown in previous studies to be misleading in some cases [1] and prone to issues such as gradient saturation [2, 3]. Furthermore, it can be hard to directly apply these methods to discrete inputs (e.g., adjacent matrix, existence of points) due to its fundamental assumption.
>
> - BernMask and BernMask-P are based on two widely used masking-based post-hoc methods. Even though these methods make it possible to directly work on discrete input space, their post-hoc nature makes them suffer from multiple fundamental issues, such as overfit $\mathcal{C}$ and underfit $\mathcal{C}_s$, and unable to provide stable/faithful interpretation results [4, 5, 6]. Moreover, these methods technically cannot well handle irregular geometric data (i.e., point cloud data with varied sizes and important geometric features) in that they propose to use sparsity-driven L2 regularization, which tends to generate small-sized $\mathcal{C}_s$ and this can be problematic as the proportion of important points can vary significantly across different GDL datasets, and that these methods provide no ways to well utilize geometric features, which makes them unable to fully explore the potential of interpretable GDL.
>
> - PointMask (ICML 2020 WHI Workshop) is adopted as a baseline because it claims a goal of providing interpretability via masking points. However, it fails to provide quantitative results to justify its interpretability, and its design seems to be with limited principles and justification, where it uses continuous Gaussian distributions to generate discrete masks, and the masks are expected to mask out points, but then it directly multiples continuous geometric features with those discrete masks to achieve so, which actually makes the masked results have specific semantic meanings in space. We also view it as our contribution to benchmark this method quantitatively with our datasets that have ground-truth interpretation labels, and our empirical results have clearly shown the inability of PointMask to provide valid interpretation results in scientific GDL, as it is the worst baseline in most cases, which may be attributed to its less principled method design.

---

> > ### Author Response · Authors · 2022-11-13
> > **Response to Reviewer Wziv (2/4)**
> >
> > - Different from all these baselines, LRI uses a fundamentally different but sound principle (i.e., the less randomness a feature can be allowed to have during training, the more important the feature is) by creating information bottlenecks (IB) in GDL tasks.
> > LRI is also trained end-to-end and can provide inherent interpretability for general GDL backbone models, which can help to generate more faithful interpretation results and do not have the issues that those post-hoc methods would have.
> > As a result, we can see from our experiments that LRI not only provides better interpretability, but also brings extra benefits due to its connection to IB (e.g., better robustness to distribution shifts and better generalization performance).
> > In addition, LRI allows to leverage different types of randomness to work on different types of inputs under a unified framework with even better interpretation performance (e.g., Bernoulli randomness on discrete features and Gaussian randomness on continuous features), which is what previous methods cannot offer. To be specific, compared to GradGeo, LRI-Gaussian performs consistently better in terms of discovering location importance and by analyzing the learned Gaussian one can even get fine-grained interpretable geometric patterns; compared to BernMask and BernMask-P, LRI-Bernoulli performs consistently better in terms of discovering existence importance and by design LRI-Bernoulli will not tend to generate small-sized $\mathcal{C}_s$ since LRI adopts information regularization instead of sparsity regularization.
> >
> > > W2: Computation overhead of LRI-Gaussian.
> >
> > We agree that computational costs would be better to be profiled. Therefore, we profile LRI-Gaussian using dataset ActsTrack and backbone DGCNN as an example. We find that constructing KNN graphs takes only a very small fraction of time. Specifically, we profile the function `run_one_batch` in our implementation (code released) using a Quadro RTX 6000 (7.5 CUDA compute capability, the same as a Geforce RTX 2060) with a batch size of $128$, and each run of this function finishes the computation of a batch of point clouds.
> >
> > As shown in the table below (the time is averaged over $10$ batch iterations and standard deviation is also reported), the construction (and reconstruction) of all KNN graphs, on average, uses $2.49 \pm 0.08$ ms and takes only $2.83$\% of the total batch time, which is even less expensive than data logging (i.e., convert tensors from GPUs to CPUs and log them locally).
> >
> > | Module      | Time (ms/batch) | Avg. Ratio (\%)  |
> > | ----------- | ------------ | ------------ |
> > | KNN graph construction      | $ 2.49 \pm 0.08 $ | $ 2.83$ |
> > | Forward computation         | $44.07 \pm 0.97 $ | $50.16$ |
> > | Backward pass       | $21.22 \pm 2.02$  | $24.15$ |
> > | Data logging        | $18.42 \pm 1.65$  | $20.97$ |
> > | Total batch time    | $87.86 \pm 1.28$  | $100$   |
> >
> > The reason for the relatively low cost of constructing KNN graphs is
> > that the construction of KNN graphs can be efficiently parallelized on GPUs. Actually, researchers in many cases are also willing to pay extra costs for better performance/interpretability in scientific GDL due to the potential significant scientific meaning these methods may bring. For example, with $n$ being the number of points in a point cloud, [7-10] all have $O(n^2)$ cost in every layer, and some higher-order models in GDL [11-13] may even have $O(n^k)$ complexity in every layer.
> >
> >
> >
> > > W3.1: Whether the prediction and interpretation gains result from more parameters and additional computations?
> >
> > We thank the reviewer for raising this point, and below we show the total number of parameters used in each method using dataset ActsTrack and backbone DGCNN as an example. One can reproduce the table using the variable `baseline` in our implementation (code released). Note that the number of parameters of GradGeo and GradCAM is just the parameters of the backbone model (i.e., DGCNN). BernMask is trained using mini-batches as well, so it introduces parameters equal to the number of points in a batch of point clouds (e.g., 100 points per cloud and batch size 128 = 12,800 extra parameters).
> >
> > | Method      | Num. of param. |
> > | ----------- | ------------ |
> > | GradGeo       | 176,163      |
> > | GradCAM       | 176,163      |
> > | BernMask      | 189,474      |
> > | BernMask-P    | 193,188      |
> > | PointMask     | 352,297      |
> > | LRI-Bernoulli | 193,188      |
> > | LRI-Gaussian  | 193,838      |

---

> > > ### Author Response · Authors · 2022-11-13
> > > **Response to Reviewer Wziv (3/4)**
> > >
> > > > W3.2: The authors claim that existing interpretation methods cannot handle irregular geometric data. However, there is limited evidence to show that the proposed methods can handle it.
> > >
> > > With what we stated for W1, the inability of previous methods when handling irregular geometric data (i.e., point cloud data with varied sizes and important geometric features) should get clear. Below we further state the evidence that LRI can handle it from two perspectives.
> > >
> > > **Technically:**
> > >
> > > - Our proposed framework LRI utilizes information regularization instead of sparsity-driven regularization (i.e., L2 norm), which makes LRI not sensitive to changes in the size of point clouds.
> > > - The design of LRI-Gaussian makes it possible to explicitly utilize the geometric features in such data.
> > > - LRI is a general framework that can be easily merged into GDL models that leverage geometric symmetry.
> > >
> > >
> > > **Empirically:**
> > >
> > > - Our experiments show the superiority of LRI on such datasets, where LRI-induced models outperform all baselines by a large margin and LRI-Gaussian achieves the best performance in most cases.
> > > - None of previous studies can provide fine-grained interpretable geometric patterns in GDL tasks, while LRI-Gaussian is the first work that can provide such results, which may have the potential to further advance scientific discovery (e.g., how would the locations of amino acids affect protein folding?).
> > >
> > > > W3.3: The authors claim that most existing methods can only provide either the existence or location importance. However, is this really a significant problem? Can the proposed LRI-Bernoulli and LRI-Gaussian work together?
> > >
> > > As we have explained the technical contributions of LRI for W1, now it should be clear how LRI is technically better than existing methods. Meanwhile, both types of importance can be tested using the same LRI framework without the need for different methods. It is also possible to make LRI-Bernoulli and LRI-Gaussian work together by utilizing the idea of ensemble methods (e.g., weighted ranking).
> > > As we have provided a unified and powerful framework, we think it can be an interesting and doable future work to let LRI-Bernoulli and LRI-Gaussian be directly trained together.
> > >
> > >
> > >
> > > > M1: The structures of $g$ and $f$ is unclear, how the $f$ take points and 3D spatial coordinates to make predictions?
> > >
> > > $f$ and $g$ can be any typical permutation equivariant GDL models as those reviewed in Sec. 3 (GDL Models), which makes LRI a general framework that can be applied to a wide range of GDL backbone models. Basically, these backbone models should take as input the point set $\mathcal{C}$, output the embeddings of points, and then MLPs will be used to make further predictions based on the learned embeddings.
> > >
> > > > M2: How is the $|\mathbf{\Sigma}_v|$ used to rank the location importance?
> > >
> > > LRI-Gaussian learns a $\mathbf{\Sigma}_v$ for every point $v$ in the point cloud, where $\mathbf{\Sigma}_v$ represents the covariance matrix of the learned Gaussian noise that can be added to the geometric features of point $v$. After training, we calculate the determinant of the learned covariance matrix, i.e., $|\mathbf{\Sigma}_v|$, and the smaller the determinant is, the more important the point is.
> > >
> > > This is because the determinant of covariance matrix $\mathbf{\Sigma}_v$ measures the entropy of Gaussian, which indicates the level of randomness of the learned Gaussian noise. When the learned $|\mathbf{\Sigma}_v|$ is small, it indicates the model tends to refuse to add large noise on point $v$ (because otherwise the cross-entropy loss will penalize too much), which implies the geometric location of point $v$ is important to the classification task. On the contrary, if the learned $|\mathbf{\Sigma}_v|$ is large, it means the model finds it to be fine to add large noise on point $v$ without hurting any cross-entropy loss, which implies the geometric location of point $v$ is less important. Therefore, we calculate the $|\mathbf{\Sigma}_v|$ of each point $v$, and use them to rank the location importance.
> > >
> > >
> > > > M3: How the eigenvectors of $|\mathbf{\Sigma}_v|$ is calculated?
> > >
> > > We actually calculate the eigenvectors of $\mathbf{\Sigma}_v$
> > > after training, where each point $v$ would have a learned $3\times3$ covariance matrix $\mathbf{\Sigma}_v$ of the Gaussian noise. Then, we apply eigendecomposition on it to analyze the learned noise. Given the small size of the matrix, the eigenvectors can be given by any eigendecomposition method, and we directly use the method implemented in Pytorch.

---

> > > > ### Author Response · Authors · 2022-11-13
> > > > **Response to Reviewer Wziv (4/4)**
> > > >
> > > > [1] Adebayo, Julius, et al. "Sanity checks for saliency maps." Advances in neural information processing systems. 2018.
> > > >
> > > > [2] Shrikumar, Avanti, Peyton Greenside, and Anshul Kundaje. "Learning important features through propagating activation differences." International conference on machine learning. 2017.
> > > >
> > > > [3] Sundararajan, Mukund, Ankur Taly, and Qiqi Yan. "Axiomatic attribution for deep networks." International conference on machine learning. 2017.
> > > >
> > > > [4] Rudin, Cynthia. "Stop explaining black box machine learning models for high stakes decisions and use interpretable models instead." Nature Machine Intelligence. 2019.
> > > >
> > > > [5] Laugel, Thibault, et al. "The dangers of post-hoc interpretability: Unjustified counterfactual explanations." arXiv preprint. 2019.
> > > >
> > > > [6] Miao, Siqi, Mia Liu, and Pan Li. "Interpretable and generalizable graph learning via stochastic attention mechanism." International Conference on Machine Learning. 2022.
> > > >
> > > > [7] Qu, Huilin, Congqiao Li, and Sitian Qian. "Particle Transformer for Jet Tagging." International Conference on Machine Learning. 2022.
> > > >
> > > > [8] Satorras, Vıctor Garcia, Emiel Hoogeboom, and Max Welling. "E (n) equivariant graph neural networks." International conference on machine learning. 2021.
> > > >
> > > > [9] Ganea, Octavian-Eugen, et al. "Independent se (3)-equivariant models for end-to-end rigid protein docking." International Conference on Learning Representations. 2021.
> > > >
> > > > [10] Wang, Yue, et al. "Dynamic graph cnn for learning on point clouds." Acm Transactions On Graphics. 2019.
> > > >
> > > > [11] Morris, Christopher, et al. "Weisfeiler and leman go neural: Higher-order graph neural networks." Proceedings of the AAAI conference on artificial intelligence. 2019.
> > > >
> > > > [12] Maron, Haggai, et al. "Provably powerful graph networks." Advances in neural information processing systems 32 (2019).
> > > >
> > > > [13] Gasteiger, Johannes, Janek Groß, and Stephan Günnemann. "Directional message passing for molecular graphs." International Conference on Learning Representations. 2020.

---

> > > > > ### Author Response · Authors · 2022-12-05
> > > > > **Thanks for your review**
> > > > >
> > > > > Dear Reviewer Wziv,
> > > > >
> > > > > As the discussion period is closing, we hope our responses provide a convincing justification for our technical contributions and address the concerns about the computation overhead and the potentially missed ablation studies. We will be glad to clarify any further questions or comments.
> > > > >
> > > > > Best,
> > > > >
> > > > > Authors

---

> > > > > ### Comment · Reviewer_Wziv · 2022-12-08
> > > > > **Some concerns are addressed while some are not.**
> > > > >
> > > > > The response addresses most concerns regarding missing details. However, the response regarding technical contributions is not well organized. The response seems to be specially tailored to answer my questions which are indeed general and fundamental concerns. There is no plan to integrate the justification or missing experiments into the final version to improve the paper. Conceptually, I appreciate the topic and idea. Technically, I still have concerns regarding incremental technical contribution and missing experiments. I would like to keep my original ranking.

---

> > > > > > ### Author Response · Authors · 2022-12-08
> > > > > > **Thanks and further response to Reviewer Wziv**
> > > > > >
> > > > > > We greatly thank the reviewer for taking the time to check our long response and provide valuable comments. We would like to further address the remaining concerns.
> > > > > >
> > > > > > We are sorry for not answering your previous questions on our technical contributions in a general and fundamental way because we assumed you were asking to compare our method with each of the previous methods separately, and that was what we did. From a general view, our new technical contributions are as follows:
> > > > > > 1. The first work well utilizes the information bottleneck principle in Geometric Deep Learning (GDL) to build interpretable models.
> > > > > > 2. We propose two different interpretation approaches for GDL models, where none of the previous studies ever looked into leveraging informative geometry features in GDL, and LRI is the first one. Those models like GNNExplainer and PGExplainer adopt some Bernoulli masks instead of Gaussian perturbation to understand continuous geometric features.
> > > > > > 3. We design to utilize multivariate Gaussian noise for geometric interpretation, where the reparameterization trick used for multivariate Gaussian is first proposed by us, and such parameterization further makes LRI the first work that can provide fine-grained geometric interpretation results.
> > > > > >
> > > > > > As we have done all these justifications and experiments in our responses, they can be incorporated into our manuscript easily, and we will be happy to incorporate them into our final manuscript if this paper gets accepted.
> > > > > >
> > > > > > We are looking forward to your re-evaluation! Sorry again for kind of misunderstanding your previous ask.
> > > > > >
> > > > > > The authors

---

> > > > > > ### Author Response · Authors · 2022-12-14
> > > > > > **Have we addressed your questions on technical contributions?**
> > > > > >
> > > > > > Dear Reviewer Wziv,
> > > > > >
> > > > > > Many thanks again for your time to check our response! We sent to you our new responses on the technical contributions from a universal perspective several days ago as below. We are sorry again that we only did one-by-one comparison which mismatched your previous ask.
> > > > > > We were wondering if this addressed your concerns and if you have any other concerns. We will greatly appreciate your time for re-evaluating our work.
> > > > > >
> > > > > > Many thanks,
> > > > > > The authors

---

### Official Review · Reviewer_vQZw · 2022-11-02

**Confidence:** 3
**Correctness:** 3
**Technical Novelty And Significance:** 2
**Empirical Novelty And Significance:** 3
**Recommendation:** 6

**Clarity, Quality, Novelty And Reproducibility:**

Frankly speaking, I am not an expert in protein analysis, and I feel this manuscript is not easy to follow.  It is welcome to conduct interdisciplinary research, but I strongly suggest the authors could highly improve its readability.

**Strength And Weaknesses:**

Strength:
1. It is interesting to study using interpretable learning mechanisms for point cloud data, especially for proteins that often consist of hundreds of amino acids.
2. It is also good to establish 4 benchmark datasets to facilitate interpretable GDL research.
3. Their experimental results show appealing properties.

Weaknesses:
1. The paper is not well-organized, and also is not easy to read.  I would suggest moving the dataset introduction part to a later section, and could better prepare a full system overview to better illustrate how the full system work.
2. According to Table 1, all 4 datasets seem to be not large-scale, and each dataset only has 2 classes.  I am not sure if that is sufficient for a classification benchmark.  The number of points in each sample seems to be not large either (i.e., the largest is 339.8), which should be much less than a normal visual point cloud scan (e.g., at least 1024 in ModelNet40).  Maybe the authors could explain more in the response.

**Summary Of The Paper:**

This work studies designing interpretable geometric deep learning via learnable randomness injection with 4 proposed datasets, ActsTrack, Tau3mu, SynMol, and PLBind.  They show that using LRI mechanisms could improve the prediction performance of the backbone models, and even could improve model generalization.

**Summary Of The Review:**

At the current stage, I think it seems to be interesting research.  I would be willing to reconsider when the other review and the author's responses are available.

---

> ### Author Response · Authors · 2022-11-13
> **Response to Reviewer vQZw (1/2)**
>
> We greatly thank Reviewer vQZw for appreciating our contributions to interpretable GDL research, the valuable suggestions on improving the readability, and supporting the acceptance of this work. We address the concerns as follows.
>
>
> > W1: The readability of the dataset description.
>
> We thank the reviewer for pointing out this concern. Because of the page limitation, we address it by adding comprehensive descriptions of the background and tasks of our datasets in Appendix B, which should help readers to better understand the settings of the proposed datasets.
>
>
>
> > W2: All 4 datasets seem to be not large-scale, and each dataset only has 2 classes. The number of points in each sample seems to be not large either, compared to a normal visual point cloud scan.
>
> We intentionally focus this work on scientific applications, because we believe model interpretability is more crucial for applications in scientific fields, where interpretable methods hold the promise to build more trustworthy models and further advance scientific discovery. Therefore, the characteristics of the datasets are determined by the nature of the corresponding scientific problems, but are not picked by us manually.
>
> Specifically, it can be a lot harder to collect labeled samples for scientific applications, compared to visual point clouds. For example, it can be extremely hard to measure molecular properties or protein-ligand binding affinities experimentally, and it can also be expensive to simulate full collision events in HEP, which greatly limits the available number of labeled samples in scientific GDL. Actually, the proposed datasets ActsTrack and Tau3mu are both filtered from one million full collision events, and SynMol and PLBind are constructed based on two of the largest datasets in the corresponding scientific domains, i.e., ZINC [1] and PDBBind [2].
>
> Similarly, the number of points of each sample in our datasets is determined by the corresponding scientific domains, where the number of points in ActsTrack and Tau3mu is decided by the underlying physics processes, and the number of points in SynMol and PLBind is decided by the number of atoms/amino acids in the measured molecules/proteins.
> For example, ZINC [1], QM9 [4], and MoleculeNet [5] are widely used molecular property prediction datasets in scientific GDL, where, on average, each molecule has 23.2 atoms in ZINC, 18.0 atoms in QM9, and 25.5 atoms in MoleculeNet-HIV. Similarly, according to [7], the median protein length (number of amino acids) is 361 in eukarya, 267 in bacteria, and 247 in archaea organisms. Likewise, datasets from HEP may have 10-100 points with an average of 30–50 points in each sample for jet tagging tasks [8, 9], and each sample may have 6-12 points for event classification tasks [10, 11]. Therefore, our proposed datasets have similar sizes of point clouds to the typical applications in scientific GDL.
>
> We agree that collecting datasets with more classes is a valuable future work to further facilitate interpretable GDL research. As this is the first work that collects datasets for this domain, we can expect more people to join and enrich the datasets in the future.
> As for the datasets proposed in this work, we formulate the tasks primarily based on their scientific motivations, and we did not intentionally constrain the number of classes. Binary classification just happens to be a quite effective formulation in scientific GDL, and this formulation can already serve the scientific purposes in many tasks [5, 6, 8].

---

> > ### Author Response · Authors · 2022-11-13
> > **Response to Reviewer vQZw (2/2)**
> >
> > [1] Irwin, John J., et al. "ZINC: a free tool to discover chemistry for biology." Journal of chemical information and modeling. 2012.
> >
> > [2] Wang, Renxiao, et al. "The PDBbind database: Collection of binding affinities for protein-ligand complexes with known three-dimensional structures." Journal of medicinal chemistry. 2004.
> >
> > [3] Wu, Zhirong, et al. "3d shapenets: A deep representation for volumetric shapes." Proceedings of the IEEE conference on computer vision and pattern recognition. 2015.
> >
> > [4] Ramakrishnan, Raghunathan, et al. "Quantum chemistry structures and properties of 134 kilo molecules." Scientific data. 2014.
> >
> > [5] Wu, Zhenqin, et al. "MoleculeNet: a benchmark for molecular machine learning." Chemical science. 2018.
> >
> > [6] Townshend, Raphael JL, et al. "Atom3d: Tasks on molecules in three dimensions." Advances in Neural Information Processing Systems. 2021.
> >
> > [7] Brocchieri, Luciano, and Samuel Karlin. "Protein length in eukaryotic and prokaryotic proteomes." Nucleic acids research. 2005.
> >
> > [8] Qu, Huilin, and Loukas Gouskos. "Jet tagging via particle clouds." Physical Review D. 2020.
> >
> > [9] Qu, Huilin, Congqiao Li, and Sitian Qian. "Particle Transformer for Jet Tagging." International Conference on Machine Learning. 2022.
> >
> > [10] Ren, Jie, Lei Wu, and Jin Min Yang. "Unveiling CP property of top-Higgs coupling with graph neural networks at the LHC." Physics Letters B. 2020.
> >
> > [11] Abdughani, Murat, et al. "Probing the triple Higgs boson coupling with machine learning at the LHC." Physical Review D. 2021.

---

> > > ### Author Response · Authors · 2022-12-05
> > > **Thanks for your review**
> > >
> > > Dear Reviewer vQZw,
> > >
> > > As the discussion period is closing, we hope our responses address your concerns about the readability of the dataset introduction and the characteristics of the proposed datasets. We will be glad to clarify any further questions or comments.
> > >
> > > Best,
> > >
> > > Authors

---

### Author Response · Authors · 2022-11-13
**Summary of Response to the Reviews**

We appreciate the valuable feedback and insightful comments from all our reviewers. All reviewers
agree the proposed research problem is important and interesting, find the proposed methods to be technically sound, and appreciate the effort of collecting four benchmark datasets from scientific applications. Reviewers vQZw, sW6V, and HwR5 also find our experiment results to be appealing and solid.

The remaining questions are mainly on the readability of the dataset description, the characteristics of the proposed datasets, and the potentially missed justification of technical contributions and ablation studies. We will address these questions in the following response to each reviewer, and we look forward to the reviewer's response.

---

### Author Response · Authors · 2022-11-19
**Looking forward to further comments from the reviewers**

Dear reviewers,

We thank all your efforts and constructive feedback in reviewing this paper. We look forward to your further comments. Thank you very much.

Best,

Authors

---

### Decision · Program_Chairs · 2023-01-20

**Decision:**

Accept: poster

**Justification For Why Not Higher Score:**

Not all of 4 reviewers champion the paper. Two of them just vote for marginal accept. Considering the quality of the current version, the AC recommend it to be accepted as a poster.

**Justification For Why Not Lower Score:**

The paper makes contribution on new method and new datasets. It is a valuable paper worthy to publish.

**Metareview: Summary, Strengths And Weaknesses:**

This paper studies interpretable geometric deep learning by designing the learnable randomness injection (LRI) mechanism.  It also proposes four datasets from real scientific applications to evaluate the proposed LRI mechanism. Experiments show that the LRI mechanisms can improve the prediction performance of the backbone models, and even improve the model generalization. All reviewers agree the research problem in the paper is important and interesting, find the proposed methods to be technically sound, and think the four collected benchmark datasets are valuable.  After the rebuttal, all 4 reviewers lean to accept the paper (particularly, 2 of them champion the paper, with each rating a score 8: accept). The rebuttal has addressed all most of the concerns raised by reviewers, including the readability of the dataset description, missed justification of technical contributions, and ablation study. The AC agrees with the judgements of the reviewers and recommends accepting the paper.   AC urges the authors to improve their paper by taking into account all the suggestions provided by the reviewers and including all additional experiment results into the revision.


**Note From Pc:**

if the above contains the word "oral" or "spotlight" please see: "oral" presentation means -> notable-top-5% and "spotlight" means -> notable-top-25%. As stated in our emails, we are disassociating presentation type from AC recommendations